# Photon quantum entanglement in the MeV regime and its application in PET imaging

D. P. Watts [1✉], J. Bordes [1], J. R. Brown [1], A. Cherlin [2], R. Newton [1], J. Allison [3,4], M. Bashkanov [1], N. Efthimiou [1,5] & N. A. Zachariou [1]

Positron Emission Tomography (PET) is a widely-used imaging modality for medical research and clinical diagnosis. Imaging of the radiotracer is obtained from the detected hit positions of the two positron annihilation photons in a detector array. The image is degraded by backgrounds from random coincidences and in-patient scatter events which require correction. In addition to the geometric information, the two annihilation photons are predicted to be produced in a quantum-entangled state, resulting in enhanced correlations between their subsequent interaction processes. To explore this, the predicted entanglement in linear polarisation for the two photons was incorporated into a simulation and tested by comparison with experimental data from a cadmium zinc telluride (CZT) PET demonstrator apparatus. Adapted apparati also enabled correlation measurements where one of the photons had undergone a prior scatter process. We show that the entangled simulation describes the measured correlations and, through simulation of a larger preclinical PET scanner, illustrate a simple method to quantify and remove the unwanted backgrounds in PET using the quantum entanglement information alone.

[1] Department of Physics, University of York, Heslington, York, UK. [2] Kromek Group, Sedgefield, County Durham, UK. [3] Geant4 Associates International Ltd., Hebden Bridge, UK. [4] Department of Physics and Astronomy, University of Manchester, Manchester, UK. [5] PET Research Centre, School of Health Sciences, University of Hull, Hull, UK. ✉email: daniel.watts@york.ac.uk

Positron emission tomography (PET) is a valuable tool for the in-vivo imaging of cellular and molecular processes. PET can provide high sensitivity and quantitative information on disease development and, more recently, therapy response. The PET information is usually complemented by purely anatomical information obtained using other imaging modalities. A typical PET study involves the administration of a radiotracer, a biologically active molecule which is labelled with a positron ($e^+$) emitting radionuclide to track metabolic activity. Subsequent $e^+e^-$ annihilation produces two 0.511 MeV $\gamma$-photons, moving in approximately opposite directions. Their subsequent detection enables a line of response (LOR) to be defined upon which the annihilation site is assumed to be located. However, in addition to this spatial information, the two annihilation $\gamma$ are predicted to be in a common entangled wavefunction which results in correlations between their interaction processes even when separated, the so-called "spooky"[1,2] action at a distance effect of quantum mechanics. Such quantum-entangled correlations have recently shown imaging benefits for optical (~eV) photons[3–5] and X-ray photons (~10 keV)[6].

For the current study we use the linear polarisation of the $\gamma$ as the experimental observable sensitive to the entangled nature of the PET photons, with visibility achieved through observation of the double Compton scattering (DCSc) process. The two $\gamma$ from (ground state) para-positronium annihilation (anti-parallel spins; $S = 0$, $S_z = 0$ where $S$ is the spin of the positronium) have orthogonal linear polarisation with an entangled wavefunction which can be expressed as

$$|\Psi\rangle = \frac{1}{\sqrt{2}}\left(|x\rangle_-|y\rangle_+ - |y\rangle_-|x\rangle_+\right), \qquad (1)$$

where $|x\rangle_-$ and $|y\rangle_-$ represent a $\gamma$ propagating along $-z$ with polarisation in $x$ and $y$, respectively. $|x\rangle_+$ and $|y\rangle_+$ have equivalent definitions but for the $+z$ direction. This entangled Bell state is the only allowed state following annihilation of ground state positronium to two photons[2,7,8].

The DCSc differential cross section of the entangled annihilation gamma in the allowed Bell state of Eq. (1), is given by[8,9]

$$\frac{d^2\sigma_{\mathrm{double}}}{d\Omega_1 d\Omega_2} = \frac{r_0^4}{16}\left(K_a(\theta_1, \theta_2) - K_b(\theta_1, \theta_2)\cdot\cos(2\Delta\phi)\right), \qquad (2)$$

where $d\Omega_{1,2}$ and $\theta_{1,2}$ are the solid angles and polar scattering angles for $\gamma_1$ and $\gamma_2$, respectively, $r_0$ is the classical electron radius,

$K_a$ and $K_b$ are kinematic factors (see Supplementary Note 1) and $\Delta\phi = \phi_1 - \phi_2$ is the relative azimuthal scattering angle (see Fig. 1 for the definition of scatter angles). The form of Eq. (2) is presented in the publication of Pryce and Ward[9] (also independently derived by Snyder et al.[7]). In these early works, consistent results were obtained when employing time-dependent perturbation theory or a Klein-Nishina[10] based approach, as outlined in the PhD thesis of Ward[8]. Subsequently, the same form has been derived in DCSc calculations in a matrix formalism[11] and employing Kraus operators[12]. The DCSc cross section is modulated by the $\cos(2\Delta\phi)$ term, resulting in an enhancement ratio ($R$) between the maximum ($\Delta\phi = \pm 90°$) and minimum ($\Delta\phi = 0°$) scattering probabilities of $R = 2.85$, achieved when both $\theta_1$ and $\theta_2$ are equal to 81.7°[7–9]. Bohm and Aharonov[2] were the first to recognise that the ($\Delta\phi$) correlations between the DCSc annihilation photons were an example of the kind of entanglement discussed by Einstein, Podolsky and Rosen[13]. They also derived[2] the upper limit for a (hypothetical) orthogonally polarised, but non-entangled (separable) state as $R = 1.63$, establishing that measured values above this limit are a witness of the entanglement[2,11,12].

Previous measurements of the $\Delta\phi$ amplitude in DCSc of positron annihilation (Pa) photons[14–21] focused on restricted kinematics of $\theta_1, \theta_2$ around maximum visibility, and yielded $R$ values well beyond the upper limit for a non-entangled state. Clear statistical agreement with the entangled theory (eq. (2)) was established (with analytic corrections for experimental aspects of the measurement). The most precise experiments were carried out by Langhoff[15] and Kasday et al.[18] giving measured enhancement factors of $R = 2.47 \pm 0.07$ and $R = 2.33 \pm 0.10$ respectively, in agreement with eq. (2) when geometrical effects are accounted for. The consistency of the measured $R$ was established[19,20] for separation distances up to 2.5 m, greatly exceeding the coherence length associated with Pa[19] (0.12 m) and larger than the diameter of a typical clinical PET scanner (0.8–1.3 m). Further discussion of the previous measurements and the appropriateness of the entanglement formalism adopted here is presented in Supplementary Note 3.

The ability of the entangled cross section (eq. (2)) to describe the $\Delta\phi$ distribution ($R$) for all previous data in positron annihilation (and for the current data—see next section) gives confidence in its implementation in PET simulation. The most useful events collected to form a PET image (true events) have a LOR which crosses the annihilation site (no interaction in the patient).

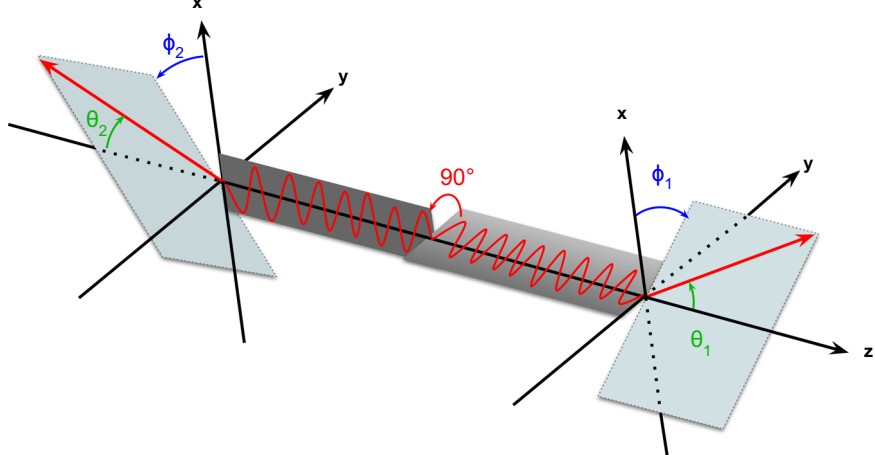

**Fig. 1 Definition of Compton scattering angles.** Schematic figure showing the definition of the scattering angles $\theta$ and $\phi$ for the double Compton scattering of the two entangled $\gamma$ photons. The direction vectors for the scattered photons are shown by the red lines, with the corresponding scatter planes indicated by the shaded rectangles. The z-axis is aligned with the $\gamma$ direction while the x-axis is defined with respect to the detector in the laboratory frame. Mutually perpendicular orientations of the polarisation vectors of the $\gamma$ are also shown (with an arbitrary orientation).

These true events are expected to maintain their quantum-entangled nature. However, these are recorded along with unwanted scatter and random backgrounds. These have LORs displaced from the annihilation site(s) which cause artefacts in the reconstructed image[22,23], decrease the signal-to-noise ratio and distort the relationship between the image intensity and the activity in the volume of interest. Scatter background arises when at least one of the two annihilation $\gamma$ scatter prior to detection. In addition to a displaced LOR, such decohering scatter will lead to entanglement loss for the subsequently detected photon pair. The scatter-to-true ratios range from ~0.2 for brain imaging to ~2 for 3D abdominal imaging[24]. The random background originates mainly from uncorrelated $\gamma$ pairs, producing LORs dispersed over the full image and which would, of course, not be in an entangled state. Random-to-true ratios range from ~0.1 (brain imaging) to more than 1[24], influenced by the detector properties (e.g the timing coincidence window for the accepted $\gamma$ photons) and the administered activity.

In this work, we implement the entangled description of the interactions of Pa photons into the comprehensive Geant4[25,26] particle simulation framework. The new simulation (QE-Geant4) is tested through a comparison with double Compton scattering of $^{22}$Na Pa gamma, measured by two state-of-the-art cadmium zinc telluride (CZT) semiconductor $\gamma$ detectors, placed in a back-to-back PET configuration. We also provide a measurement of the diminished DCSc correlation for the case where one photon in the pair has scattered prior to measurement, achieved using a modified (non-back-to-back) setup. The benchmarked QE-Geant4 simulation is used to model a preclinical PET imaging apparatus, formed from an array of CZT detectors. This quantum-entangled PET (QE-PET) simulation indicates a spatially resolved extraction of the shape and magnitude of scatter and random contributions to PET images is achievable, using only the entanglement information contained in the data and information from the QE-Geant4 simulation.

## Results

**Comparison of simulated and measured $\Delta\phi$ distributions**. We implemented the entangled description of the interactions of Pa photons into the comprehensive Geant4[25,26] particle simulation framework (see methods), which enables full account of detector geometry, experimental resolutions and backgrounds. The simulation (QE-Geant4) was tested by comparing to measurements of DCSc of Pa photons in a PET-demonstrator apparatus. The system was developed by the Kromek Group based on the DMatrix detector system[27] and comprised two 10 mm cubic semiconducting CZT crystals, placed back-to-back and separated by 87 mm. A segmented anode divided each crystal into 121 0.8 × 0.8 mm$^2$ pixels, with depth information accessible from the anode drift time[28]. A 170 kBq $^{22}$Na source, comprised of a 1 mm diameter active bead housed in a thin plastic housing, was placed equidistant from each crystal face providing a source of positron annihilation $\gamma$.

Events in which the two annihilation $\gamma$ undergo DCSc in the CZT were identified from the yield having two hit clusters in each head (see methods). The normalised (see caption) coincidence count rate for DCSc as a function of $\Delta\phi$ for the event sample is shown as the data points in Fig. 2. A strong $\cos(2\Delta\phi)$ modulation is seen with a measured enhancement factor of $R = 1.85 \pm 0.04$ for the event yield selected by the analysis cuts viz. polar scatter angles $70° \leq \theta \leq 110°$ and summed energy $480-530$ keV. Employing a tighter restriction on polar scattering angles ($\theta_{1,2} = 93-103°$) gave a measured enhancement factor $R = 1.95 \pm 0.07$.

To directly compare the QE-Geant4 simulation with experimental data the predicted energy deposits in the CZT detector

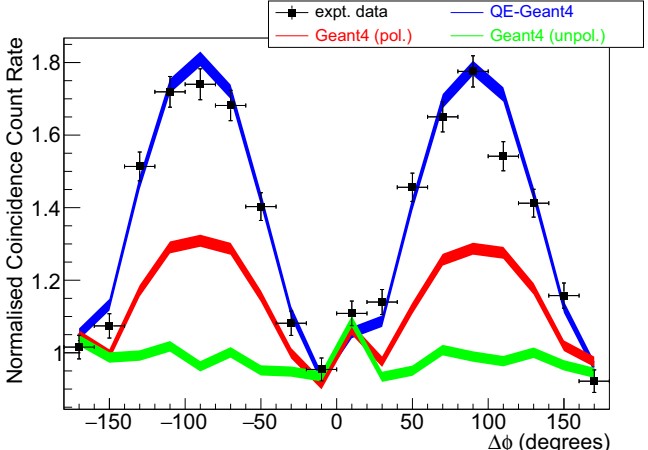

**Fig. 2 Comparison of experimental and simulated scattering probabilities.** Black data points show the experimentally determined (normalised) coincidence count rate for double Compton scattering as a function of the azimuthal difference in scatter angles ($\Delta\phi$), obtained using the Cadmium Zinc Telluride (CZT) detector apparatus (see text and Fig. 8). The events analysed were for polar scatter angles $70° \leq \theta \leq 110°$ and a summed energy of the two clusters in each CZT detector in the range 480–530 keV. The statistical error (standard deviation) calculated from the number of events in each bin is shown by the associated error bars on the points. The prediction from the quantum-entangled simulation (QE-Geant4) is shown as the blue line. Non-entangled predictions for orthogonally polarised $\gamma$ pairs, Geant4(pol), are shown as the red line. The prediction for unpolarised annihilation $\gamma$, Geant4(unpol), is shown by the green line. The standard deviation statistical uncertainties for the simulations, calculated from the yield of simulated events achieved in each bin, are indicated by the line widths. For comparison of the $\Delta\phi$ distributions, the experimental and simulated data are normalised to unity for the data around the minima at $\Delta\phi = 0°$ and $\pm 180°$, specifically the mean yields from bins 1, 9, 10 and 18.

heads were matched to the experimentally determined CZT resolutions (see methods) and then passed through the same data analysis code and cuts as the experimental data. The predictions from the QE-Geant4 simulation are shown by the blue line in Fig. 2 and show a good agreement with the measured $\Delta\phi$ distribution ($\chi^2/\nu = 1.87$). The agreement provides a validation of the QE-Geant4 simulation for describing DCSc and confirms that the entangled theory (Eq. (2)) is consistent with the experimental data (the agreement on a bin-by-bin basis is presented in Supplementary Note 4).

To better quantify the predicted effects of entanglement on the measured $\Delta\phi$ amplitude, a further simulation of (hypothetical) non-entangled, but orthogonally polarised, annihilation $\gamma$ is also presented (red line). The DCSc for this case is modelled by the standard Geant4 polarised Compton scattering classes (the consistency between standard Geant4 and the theoretical predictions[2] for such a (hypothetical) non-entangled state is shown in Supplementary Note 3). The size of the predicted $\Delta\phi$ modulation is clearly reduced if entanglement is neglected, and the predictions are in clear disagreement with the experimental data. We remark that earlier standard Geant4 simulations of polarisation effects in PET[29,30], could be further developed with this QE-Geant4 framework.

A further simulation for unpolarised annihilation $\gamma$ was also carried out (green line). The intrinsic $R$ for such events is equal to unity, in agreement with analytical calculations for mixed states in Bohm and Aharonov[2] and Caradonna et al.[11]. The predicted $\Delta\phi$ distribution for unpolarised events in the detector acceptance

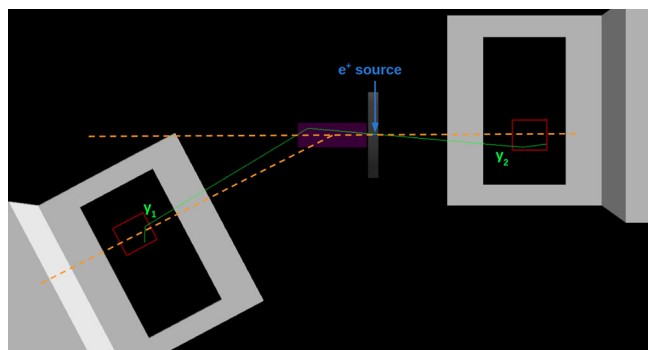

**Fig. 3 Geant4 visualisation of the setup for the scattering measurement.**
The cadmium zinc telluride (CZT) crystals (red) are shown along with their
support structures (grey) and the nylon scattering medium (purple).
The event topology for a typical scatter event is shown by the solid lines.
The initial back-to-back trajectory of the two annihilation $\gamma$ from a single
positron annihilation event can be seen as the green lines originating at
the source position. One of the photons Compton scatters in the nylon
scattering medium (purple). The subsequent Compton scatters of both
$\gamma$ within the CZT crystals, from which the $\Delta\phi$ correlation is obtained, are
evident from the kinks in the photon trajectories.

(green line) is rather uniform albeit with a small acceptance-
related enhancement near to the centre of the distribution, which
is also evidenced in the experimental data, and appears well
modelled by the simulation. It is clear that the measured $\Delta\phi$
distribution only has modest influence from detector acceptance
effects.

The results in Fig. 2, and previous measurements in more
limited kinematics from a range of different Pa sources[2,7,9,14–21],
show that the azimuthal correlation of the Compton scatter
planes in DCSc of Pa photons is in agreement with the entangled
theory (Eq. (2)) and has a measured enhancement factor ($R$)
beyond the upper limit of a separable non-entangled state.

**Investigation of entanglement loss**. We remark that the data in
Fig. 2 is dominated by photons for which their first interaction is
the (linear polarisation analysing) Compton scatter reaction in
the CZT detectors. We are not aware of any previous measure-
ment of the $\Delta\phi$ correlation for photons which have undergone an
identified decohering process prior to the measurement.

To achieve this we adapted the experimental setup as shown in
Fig. 3. Events where one of the photons has undergone a
Compton scattering process prior to measurement of the $\Delta\phi$
correlation (prior-scatter events) were obtained by inclusion
of a scattering medium (nylon) in the path of one of the
annihilation $\gamma$, with the corresponding CZT module rotated
through 33° relative to the centre of the scatterer. The measured
energy of the scattered $\gamma$ in the CZT (obtained from the sum of
the two energy deposits) matched that expected from the reaction
kinematics (~440 keV) and was well separated from backgrounds.

The DCSc $\Delta\phi$ distributions measured by the CZT detectors for
such prior-scatter events are shown by the black data points in
Fig. 4. For comparison, the red data points (red line) show the
measured (simulated) $\Delta\phi$ correlations for back-to-back unscat-
tered $\gamma$ respectively, with the same binning as used for the prior-
scatter data. The measured $\Delta\phi$ correlation for the prior-scatter
events is clearly diminished compared to the unscattered case.
The QE-Geant4 prediction (blue line) also exhibits a diminished
$\Delta\phi$ correlation, in statistical agreement with the experimental
data. In the QE-Geant4 modelling (see Methods) a complete loss
of entanglement is assumed following the first DCSc (predomi-
nantly in nylon and CZT for this event sample) and any

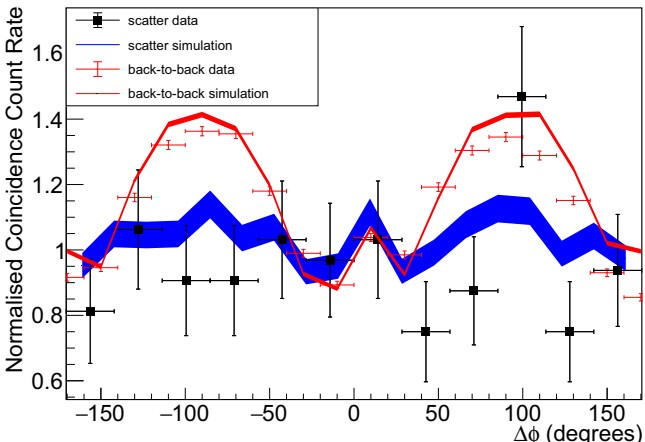

**Fig. 4 $\Delta\phi$ distributions for events with a prior-scatter process.** The black
data points were obtained with the setup of Fig. 3. They show the measured
$\Delta\phi$ distribution when one of the annihilation $\gamma$ has scattered through a polar
angle of ~33° prior to the detection in the CZT. A selection of polar scatter
angles (within the CZT) in the range 60°≤$\theta$≤140° is applied. The horizontal
error bars show the bin width and the vertical error bars show the statistical
error (standard deviation) calculated from the number of events in each
bin. For comparison the red data points show the measured $\Delta\phi$ distribution
in a back-to-back configuration (without scatterer), using an identical polar
scatter angle range for the events in the CZT detectors (error bar
definitions as for the black data). Quantum-entangled Geant4 (QE-Geant4)
predictions for the setup of Fig. 3 are shown as the blue line. The red line
shows the QE-Geant4 predictions for the back-to-back configuration
(without scatterer). All simulations employ a selection of polar scatter
angles (within the CZT) matching those applied to the data (Note that in
order to increase the event yield a wider polar angle acceptance was
employed than in Fig. 2). The standard deviation statistical uncertainties for
the simulations, calculated from the yield of simulated events achieved in
each bin, are indicated by the line widths.

subsequent gamma interactions are modelled as for polarised,
independent $\gamma$ (i.e. a separable state). As far as we are aware the
world's current data on this process for positron annihilation is
contained in Fig. 4. A future measurement programme with
improved statistical accuracy and a wider range of scattering
kinematics would be a clear next step for the field. Such
measurements (although planned) are currently out of reach of
the small PET demonstrator system used here (the data in Fig. 4
required over a month of data taking).

**Quantum-entangled PET**. The ability of QE-Geant4 to accu-
rately describe the observed $\Delta\phi$ correlations in DCSc offers
possibilities to separate the true (assumed entangled) PET events
from backgrounds of scatter and random events. The method
exploits the differences in the $\Delta\phi$ correlations to identify their
relative contribution to the image on a statistical basis (note the
correlations do not offer the possibility to identify contributions
on an event-by-event basis). To illustrate this potential, PET
images were produced from a QE-Geant4 simulation of a pre-
clinical PET acquisition. A scanner composed of four rings of
CZT (with the same crystal size as the demonstrator) was defined,
along with a standard preclinical mouse phantom NEMA-NU4
(National Electrical Manufacturers Association)[31] (Fig. 5). The
phantom consisted of a cylinder of tissue equivalent poly(methyl
methacrylate) (PMMA) with five capillaries (1–5 mm in dia-
meter) filled with a solution of water and an $e^+$ source.

To study the QE-PET imaging benefits, two-dimensional PET
images were reconstructed from the simulated data using a simple

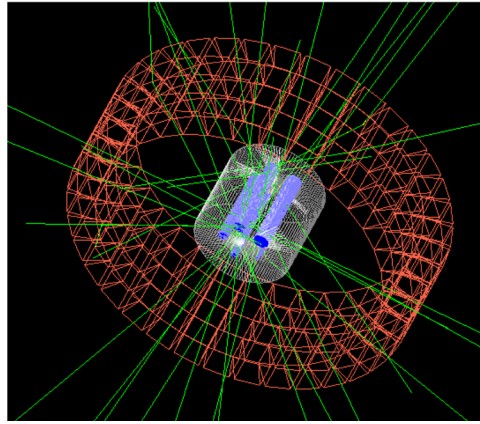

**Fig. 5 Geant4 visualisation of the simulated preclinical PET scanner.** The simulated scanner consisted of four rings of 10 mm cubic cadmium zinc telluride detector crystals (orange). The front faces of the crystals were located 49.1mm from the centre of the scanner. An industry standard NEMA-NU4 phantom was positioned in the centre of the simulated scanner. It consisted of a cylinder of PMMA (white) within which was embedded five capillaries (1-5-mm diameter) filled with a solution containing liquid water mixed with $e^+$ source (blue). The simulated trajectories of 20 pairs of annihilation $\gamma$ emanating from the phantom are shown by the green lines.

filtered back-projection (FBP) algorithm[32] and are shown in Figs. 6a and 7a, corresponding to simulations with scatter or random backgrounds respectively. Only coincidences where both $\gamma$ have polar scattering angles in the CZT within $67° \leq \theta \leq 97°$ were retained, as a compromise between keeping the enhancement ratio high and maintaining good statistics. The FBP images clearly exhibit the structure of the NEMA-NU4 phantom, showing activity from the 5 capillaries. The intensity profiles for a region of interest crossing the two lowest capillaries (seen in yellow in Figs. 6a, 7a) were extracted for two bins of $|\Delta\phi|$, $0° \leq |\Delta\phi| \leq 20°$ and $80° \leq |\Delta\phi| \leq 100°$. The bins correspond (respectively) to regions where low and high fractions of true events are expected (cf. Fig. 2). The profile for the true events can be obtained using a simple subtraction of the profiles from each $|\Delta\phi|$ bin, scaled with factors obtained from the QE-Geant4 simulation (see Supplementary Note 5). The results of this subtraction for data sets with scatter (random) backgrounds are shown by the red lines in Fig. 6d (Fig. 7d). Good agreement is observed with the "actual" true profiles (blue lines), which are reconstructed exclusively from the true coincidences identified using a priori knowledge from the simulations. An equivalent methodology can be used to extract the scatter and random distributions in isolation (Figs. 6c, 7c). The overall magnitude and shape of the QE-PET and "actual" profiles are in good agreement for both scatter and random background scenarios. The fluctuations in the extracted profile (more prominent in Fig. 6c) are not statistical in nature and appear to be caused by artefacts in the FBP imaging produced by incomplete acceptance of the array[33]. In each case, a 4th order polynomial fit to the profile is additionally included to enable the average trend to be compared with the "actual" distribution.

Even this simple illustrative method, which only uses a fraction of the available data from two of the $\Delta\phi$ bins, along with information from the entangled simulation, indicates the feasibility of quantitative assessment of both the shape and magnitude of the image backgrounds with QE-PET. Further QE-PET analysis of the extracted profiles through the middle capillaries of the phantom is presented in Supplementary Fig. 5.

## Discussion

We should remark that the information from QE-PET, as illustrated above, would be obtained in addition to the single-pixel yield routinely analysed to produce PET images. The QE-PET derived information on the backgrounds would equally well apply to these standard PET events. The results (Figs. 6, 7) were obtained from a simulation of $10^{12}e^+e^-$ annihilations, which is of the same order of magnitude as the radiotracer cumulated activity in a patient during a typical clinical PET scan (a few 100 MBq activity integrated over an acquisition time of 30 minutes). The accuracy in the extracted background profiles is therefore indicative of what may be achieved in a PET scan for this simulated detector geometry. We should also note that these proof-of-principle results employed restrictive $\theta$ and $\Delta\phi$ cuts, thereby only using a fraction of the available data. In future work, these cuts will be optimised to further improve the achievable accuracy by accounting for the interplay between enhancement magnitude and event yield, as explored previously[30,34]. However, ultimately we view the optimal use of the information would be within the framework of more sophisticated imaging methodologies such as the forward model of a MLEM (maximum-likelihood expectation-maximization)[35] image reconstruction algorithm. Currently such approaches model scatter backgrounds employing either: scatter simulation algorithms[36,37], which require analytical modelling of the scanner; or CPU intensive Monte Carlo methods. Both approaches require detailed anatomical information from a computed tomography (CT) scan and rely on estimates of the underlying activity biodistribution, which is a priori unknown and necessitates iterative approaches[23,38,39]. Implementation of the QE-PET information in such iterative imaging methods[40] would be an important next step. Time-of-flight (TOF) PET methodologies using fast photon detectors have recently been explored to address backgrounds in PET[41]. The QE-PET method can be employed in parallel with TOF information where available. However, for systems where such timing information is unavailable or of insufficient resolution (e.g. semiconductor detectors as studied here or compact PET), then QE-PET would offer unique opportunities. The PET study presented here corresponds to a preclinical PET scanner with a small-animal equivalent phantom. The scatter probability from the small-animal phantom, obtained from the QE-Geant4 simulation, is 15% providing a test of the QE-PET technique for the challenging scenario of small scatter backgrounds. Human PET, as referenced in the introduction, provides larger scatter contributions of up to 67% and should be more amenable to the method proposed.

In summary, we have simulated the predicted effects of quantum entanglement in the interaction of positron annihilation photons with matter, building on the Geant4 simulation framework. The QE-Geant4 simulation predictions were validated by comparison with precision experimental data on double Compton scattering of positron annihilation photons, obtained with a CZT pixelated semiconductor PET demonstrator apparatus. The inclusion of quantum entanglement in the simulation for the reaction process gave a good description of the measured correlation between the Compton scatter planes, while predictions based on a (hypothetical) non-entangled state could not describe this correlation. As well as underpinning the quantum-entangled PET (QE-PET) developments, the simulation will enable improved accuracy in any future simulations of standard PET for medical imaging, medical research or industrial applications. Additionally, the framework offers possibilities for further fundamental tests of entanglement at the MeV scale. A modified apparatus, incorporating an additional scattering medium, enabled a first measurement of the Compton scatter plane correlation when one of the (initially entangled) annihilation photons underwent a Compton scatter process prior to measurement.

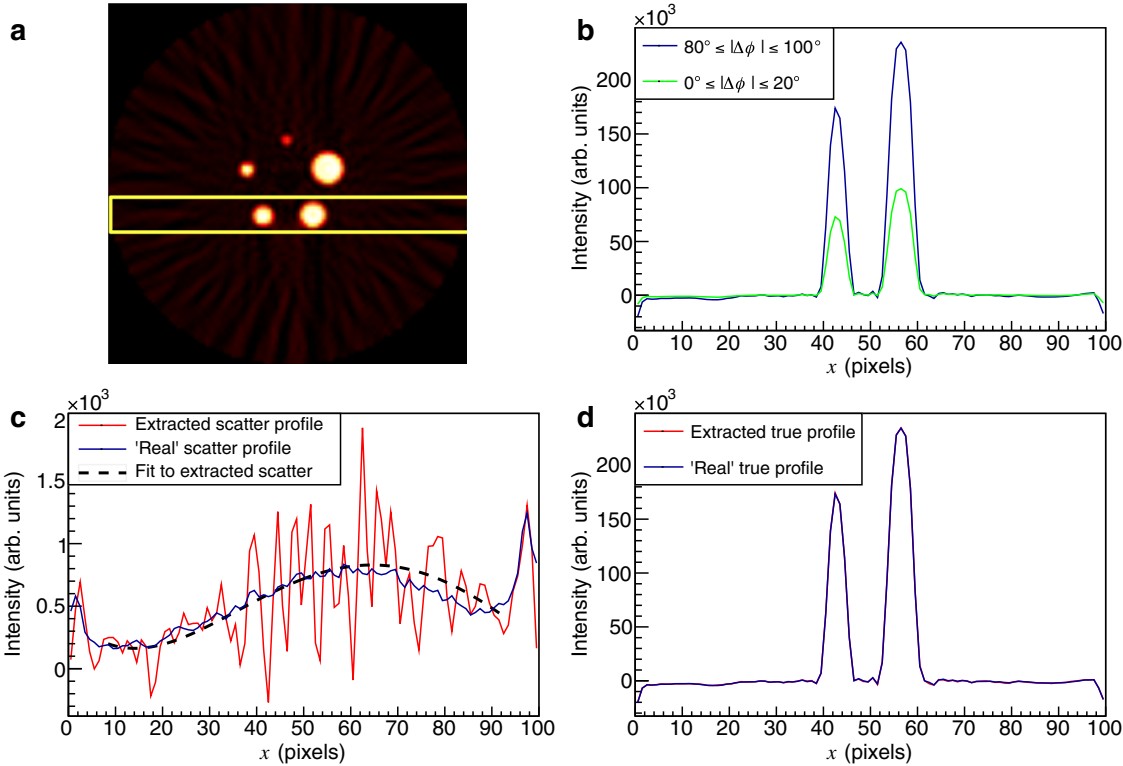

**Fig. 6 Extraction of true and scatter contributions. a** Filtered back projection two-dimensional PET image of the NEMA-NU4 phantom for true events with a scatter background. **b** Intensity profiles through the region indicated by the yellow rectangle for different $\Delta\phi$ cuts, i.e. $0° \leq |\Delta\phi| \leq 20°$ (green), and $80° \leq |\Delta\phi| \leq 100°$ (blue). **c** Extracted scatter background profile using quantum-entangled PET (QE-PET), obtained from a scaled subtraction of the two $\Delta\phi$ cut profiles (red line) (see text). The blue line shows the profile from the "actual" scatter events in isolation using a priori information from the simulation. The dashed line is a 4th order polynomial fit to the extracted scatter profile. **d** Profile extracted for true events with QE-PET (red line) compared to the profile of "actual" true events (blue line).

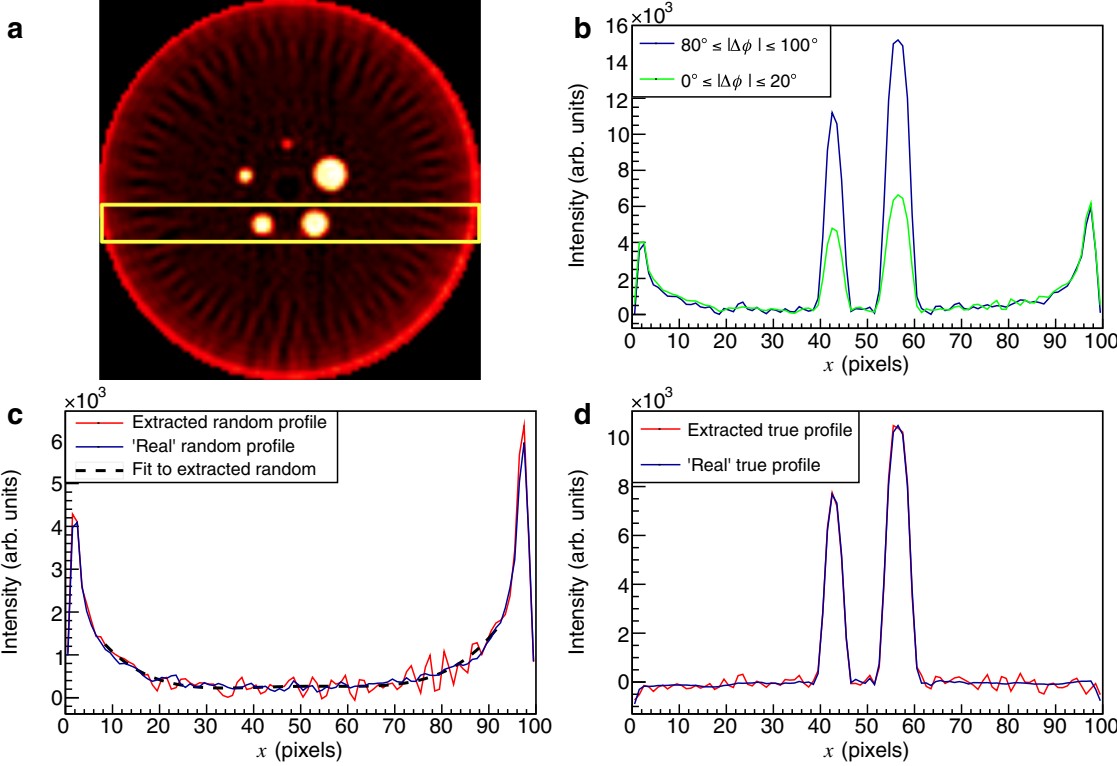

**Fig. 7 Extraction of true and random contributions.** A simulated scenario corresponding to true events with a random background. The panels **a**–**d** have the same analysis cuts as described in Fig. 6.

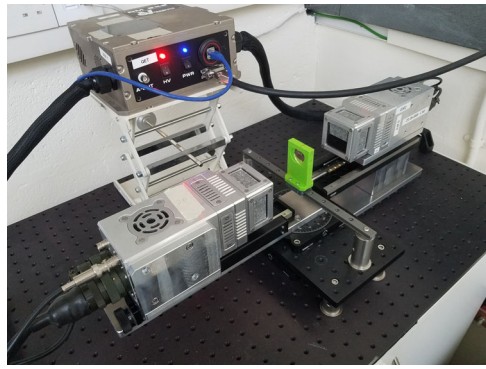

**Fig. 8 Photo of the experimental setup.** The layout of the experimental apparatus showing the Aluminium CZT detector head casings and radioactive source holder (green). The heads are mounted on a rail system comprising both linear and rotational stages.

The experimental data are in statistical agreement with simulation predictions which assume the collapse of the entangled wavefunction following the scatter process. The QE-Geant4 simulation was also used to model a CZT based preclinical scanner and obtain a simulation study of QE-PET imaging. A method to quantify and remove backgrounds from both scatter and random coincidences is presented, suggesting quantum-entangled PET provides possibilities to address key challenges for next-generation PET imaging.

## Methods

**Implementation of entanglement in Geant4 simulation.** We incorporated entanglement (as described by Eq. (2)) into Geant4, by developing routines which enable communication between the individual particle-tracking processes to reproduce the scattering cross sections for entangled $\gamma$ pairs. The simulation framework, which we refer to as QE-Geant4, will be included in future releases of the code. For the current study QE-Geant4 was implemented into Geant4 version 10.5. The non-entangled and unpolarised predictions were obtained using an unmodified version 10.5.

The implementation of the entanglement is as follows. The standard physics routines in Geant4 for modelling Compton scattering of polarised gamma use the polarised Klein-Nishina theory embedded within the "Livermore" physics package[42,43]. In QE-Geant4 this modelling is adopted for all photon tracking processes, other than the first DCSc for which the entangled formalism is applied according to Eq. (2). The two Pa gammas ($\gamma_1$, $\gamma_2$) are processed sequentially: $\gamma_1$ is tracked until it is destroyed using the standard Geant4 processes, but the kinematics of its first Compton scatter are stored and made accessible to the tracking processes of other particles. For $\gamma_2$ the modelling of its (first) Compton scatter is chosen according to the distribution of Eq. (2), using the stored Compton scatter information from $\gamma_1$. Any subsequent Compton scatter processes for either gamma revert to standard Geant4, i.e. the first DCSc process is assumed to completely collapse the entangled state of Eq. (1), with any subsequent interactions of the $\gamma$ modelled as independent and separable photons. Supplementary Fig. 6 shows the entanglement implementation (discussed above) as a flow diagram.

A check of the QE-Geant4 simulation predictions through a comparison with the underlying analytic entangled theory[7,9] is presented in Supplementary note 2. The consistency between the standard Geant4 simulation predictions and the analytic theory for DCSc of a (hypothetical) non-entangled state[2] is also presented.

**The CZT PET demonstrator apparatus.** A photo of the experimental setup is shown in Fig. 8. The properties of the CZT directly relevant to the analysis are included in the main text. The readout of the anode and cathode signals from the CZT employed a bespoke Application Specific Integrated Circuit (ASIC). The coincidence timing window for the system was set to 1 $\mu$s due to the charge sweep-out time of a few tenths of $\mu$s.

**Identification of double Compton scatter events.** Measurements of $\Delta\phi$ were accessible for events where both annihilation $\gamma$ interacted through a Compton scattering and a subsequent photoelectric absorption. Such events produced, in both CZT modules, two clusters of pixels with a total energy in the range 480–530 keV. Charge sharing events, produced by a single hit between adjacent pixels, were rejected by requiring a gap of at least one pixel between clusters. The polar scattering angle was determined using the Compton scattering formula, assuming the largest energy signal as the first interaction. Simulations indicate that this

assignment will be correct in ~60% of cases. Incorrect assignment will result in a calculated polar scattering angle $\theta_{calc.} = 180° - \theta_{real}$, but not affecting $\Delta\phi$ due to the symmetry about 0°. To enhance the amplitude of the $\cos(2\Delta\phi)$ distribution (Eq. (2)), coincidences were only retained if the polar scattering angle was within $70° \leq \theta \leq 110°$. The azimuthal scattering angle was simply determined from the energy-weighted centre of gravity of each cluster and only one- or two-pixel clusters were considered. The azimuthal angular resolution depended on the distance between the clusters. It ranged from 2.9° (for the most distant pixels) to 20.4° for the closest pixels considered in this analysis. The energy and $\theta$ cuts reduce the data to 29.2% and 5.4% of the total two-cluster events respectively, and when combined retain 2.6% of the total yield.

We remark that future higher statistics data would enable the selection of a subset of Compton scatter events with improved azimuthal angular resolution (larger inter-cluster separations) than the data included in Fig. 2. This would be expected to increase the visible enhancement factor $R$ accessible from the CZT apparatus.

**Implementing experimental resolutions in the simulation.** The CZT experimental setup was simulated with QE-Geant4. To allow a direct comparison of simulated and experimental data the simulated energy deposits were matched to the CZT experimental detector response by accounting for effects of charge transport, diffusion and self-repulsion[44], and smeared to match the experimental energy resolution of each detector module (3.8 and 5.3% FWHM at 662 keV). The location of the QE-Geant4 predicted energy deposits were convoluted with the position resolution of the CZT detectors, with coordinates of the simulated energy deposits smeared according to a Gaussian distribution with $\sigma = \frac{0.8\,mm}{\sqrt{12}}$. The resulting simulated data were then passed through the same analysis code and cuts as for the experimental data.

**Details of the PET image extraction method.** A scanner, comprising of an array of the CZT detectors used for the PET-demonstrator apparatus (see Fig. 5) was implemented in QE-Geant4. The predicted energy deposits were matched to the experimentally observed resolutions of the CZT detectors as described above. The simulated QE-PET event data were stored in a list-mode file, including the $\theta_{1,2}$ and $\Delta\phi$ information for each event. The PET events were sorted into three groupings according to their event type: the first group had no condition on $|\Delta\phi|$, the second and the third contained only events having $0° \leq |\Delta\phi| \leq 20°$ or $80° \leq |\Delta\phi| \leq 100°$, respectively.

Image reconstruction was then performed for each data group, using the Filtered Back Projection (FBP) methodology implemented in GAMOS (Geant4-based Architecture for Medicine-Oriented Simulations)[45]. For each data group, the PET events were histogrammed into sinograms with the "lm2pd" utility. Images were reconstructed with the implementation of the single-slice rebinning FBP (SSRB-FBP2D) algorithm[46]. A ramp filter was applied. Pixel size was set to $0.6 \times 0.6\,mm^2$. Images were processed with the NucMed plugin of ImageJ[47].

We note that the fluctuations in the extracted profiles (evident in Figs. 6c, 7c) have general features which are mirrored in the (statistically independent) profiles extracted for each $\Delta\phi$ bin (before they are subtracted to extract the profile). We therefore take these fluctuations not to be dominated by statistical effects. Such fluctuations have been observed previously in FBP[33] and post-processing methodologies have been attempted[48–50]. Note that for the initial assessments presented in the current work, no additional processing of the FBP image was applied.

## Data availability
The analysed experimental data presented in the manuscript would be made available upon request. The data from the PET imaging analysis will also be available upon request.

## Code availability
The classes developed for the simulation of the quantum-entangled $\gamma$ will be accessible to the community in a future public release of Geant4 open source code, along with appropriate documentation for its use.

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

## Acknowledgements

The authors acknowledge input from D. Jenkins to the manuscript. Simulation work was undertaken on the Viking Cluster, which is a high performance computing facility provided by the University of York. We are grateful for computational support from the University of York High Performance Computing service, Viking and the Research Computing team. The work has been supported by funding from Innovate UK EP/P034276/1 and the UK Science and technology Facilities Council (STFC) ST/K002937/1

## Author contributions

D.P.W. conceived the original concept, supervised the project, and wrote the initial manuscript. Initial modifications to Geant4 to include entanglement were performed by D.P.W. with final implementation into the simulation source code done by J.A. Further Geant4 development and simulations were carried out by J.B. and R.N. Simulation outputs were modified to compare to experiment by A.C. and J.R.B. The detector system was developed and calibrated by A.C., with data acquisition and analysis performed by A.C., J.R.B., and R.N. Image reconstruction was performed by J.B. and N.E. Development of the procedure and implementation to extract the profiles carried out by J.R.B. and D.P.W. M.B., N.E. and N.Z. contributed to the development of the project and supervision of students. All authors participated in the discussion of the results and contributed to the manuscript.

## Competing interests

The concept of quantum-entangled PET imaging is filed as patent PCT/GB2015/053786 by D.P. Watts. Kromek Group has invested in the exploitation of the method with the CZT detector apparatus.
