## [Peer Review File · Nature Communications]

Reviewers' Comments:

Reviewer #1:

Remarks to the Author:

This paper describes experimental work and simulations that demonstrate the theoretically predicted angular correlation of photon polarization of annihilation gamma rays from positronium decay due to quantum entanglement. It is claimed, correctly to the best of my knowledge, that this is first time this has been demonstrated. This work is presented in the context of PET medical imaging and additional simulations indicate a potential application whereby the angular correlation can be used to distinguish true coincidence events from scattered and random events in PET. Despite the uncertainty as to whether or not this could one day become a practically useful technique, this is clearly a very exciting result and will be of great interest, possibly well beyond those working in the field(s) . There are various issues that should be addressed in manuscript. The manuscript appears technically sound, and the emphasis is clearly on the main result, however many of the descriptions on both results and context are quite brief and could be made clearer/enhanced by providing more technical information, also there are a few technical issues that should be addressed. These are listed below

1. line 6 - refs 2 and 3 are quite dated - please provide more up to date ones

2. line 14- radioisotope>radionuclide

3. line 24 - ref 3 is only about scatter;

4. As stated, scatter and randoms can make up a large fraction of the collected data in PET. There are established effective ways of correcting for the systematic errors they introduce, and the most significant detrimental they have is in reducing the final image SNR (characterized by 'noise equivalent count rate'). It is indicated how the additional information obtainable with this method allow true and scatter/randoms distributions to be determined, but as there does not seem to be the potential to discriminate on an event by event basis (only probabilities are obtained) and explicitly reject scatter/randoms, the effect on SNR will be limited. Rather than performing simple scaling/subtraction-type corrections, the additional information would now be exploited via a more sophisticated reconstruction procedure, for example incorporating the process into the forward model. I appreciate the proposed method is primarily for demonstration purposes, but these issues should be referred to in the discussion

5. line 51- there has been a lot of recent discussion in the literature on this topic that is not referred to here (ie and should be) eg. Hiesmayr & Moskal Scientific Reports | (2019) 9:8166 | <https://doi.org/10.1038/s41598-019-44570-z>; Caradonna et al 2019 J. Phys. Commun. 3 105005

6. line 75 - it is appreciated that this detector system is intended primarily as a demonstrator, however there are only limited systems that could measure the azimuthal angles in a PET configuration so it would be useful to provide some other relevant parameters of the CZT system - notably what is the temporal resolution, what is the angular (azimuthal)resolution for the scattered gamma direction , can 1st v 2nd hits be reliably discriminated. Additionally most current generation PET scanners acquire time-of-flight information - the implications of this should be commented on. This could be in the discussion.

7. For readers unfamiliar with this topic a diagram indicating the detector setup and the various angles referred to would be useful

8. In the description on page 3, describe also the correlation expected if entanglement is not included, ie to provide context for the red line in fig 1. Also refer to the residual correlation after scattering

9. line 71- comment on the reduction in total true coincidence sensitivity resulting from the

restriction on theta to around 82 degrees

10. Fig 1 - this figure is the key result of the paper and should better presented. In particular - what do the error bars signify; how is the normalisation done exactly ; what is the measured 0-90 difference of the theoretical 2.85, and what is the error on that (if there is agreement, then ideally there should be statistical measure, the text just states 'good agreement') ?. What is the significance of this in the context of recent publications , eg Caradonna et al in 2019 (above) state, 'to date, no Compton scattering experiment of annihilation photons is known that has directly confirmed the predicted value of $R=2.85'$

11. line 100 - a diagram would be helpful here

12. line 107 - analysis cuts ARE SHOWN

13. fig 2 - Please provide some interpretation of what is seen here. Presumably the peaks in simulated data at +/- 90 degrees are assumed to be real ? ; move the caption so it doesn't cover the data.

14. line 140 - restate here the theta acceptance criteria

15. line 147 - I appreciate this is a simple demonstration of the potential of the technique , but please comment on the constancy of the scaling factors and how they would be affected by the residual correlated polarization in the scatter for different source and attenuator distributions

16. Fig 4 - Clarify the procedure here and in the text - the scaling factors refer to the fraction of T, S in the various phi windows in the raw data. They are not applied to the profiles in the reconstructed images shown in the figure (ie if that is the case, then it is incorrect)

17. Fig 4 C - This profile does not resemble what a scatter profile typically (ie always!) looks like (either through the image or through the raw data). In general the scatter profile will not reflect detail of the emission distribution (ie which 4c does). For example see the typical profiles for a similar object/scanner geometry shown in fig 7 of Bentourkia et al Computerized Medical Imaging and Graphics 33 (2009) 477-488. Possibly this is a result of the energy spectrum of the detected scatter for this unusual detector set up, but it requires an explanation

18. line 194 - This is an exciting result but the discussion should also comment on the large price in sensitivity that is paid due to the various selection criteria applied to the data, in order to extract the angular correlation information. It would be useful to refer to Moskal et al Eur. Phys. J. C (2018) 78:970 <https://doi.org/10.1140/epjc/s10052-018-6461-1>. where these issues are discussed in detail .

19. line 293 - CT scans aren't required/performed to determine scatter (well not additional ones anyway)

Reviewer #2:

The authors claim to have carried out "the first simulated study of quantum entanglement effects in PET imaging and validated through comparison with experimental data from their demonstrator". They further claim that their MonteCarlo programme based on the standard Geant-4 "gives the first experimental constraint of entanglement breaking" and show its usefulness by "removing scatter and random backgrounds".

The draft has several shortcomings, omissions and serious problems that I will list point by point in the following:

A) The idea of using the inherent correlation of two gamma quanta to remove background is not new. It has been tackled by e.g.

[Positron emission tomography coincidence detection with photon polarization correlation

Aimee L. McNamara, Kinwah Wua, David Boardman, Mark I. Reinhard and Zdenka
Kuncic: <https://www.spiedigitallibrary.org/conference-proceedings-of-spie/8668/86681U/Positron-emission-tomography-coincidence-detection-with-photon-polarization-correlation/10.1117/12.2007794.short>

or reference 16 cited in the draft.

In addition, there are no quantitative statements regarding a possible improvement of scatter fraction reduction. There is also no discussion about the decrease of the PET sensitivity due to the measurements of scattered photons and the reduction of events due to the limitation of the relative angle between scattering planes to a range close to 90 degrees.

B) The authors claim that for "the first time" via Geant-4 the entanglement has been taken into account, which they denoted as QE-Geant4 (QE...quantum entangled). First, the programme is not published with the paper, they announce that it will be published (with their company?). Secondly, an MC can only simulate quantum mechanical effects (it is per se a classical computer programme) and is used in all standard experiments where entanglement plays a role such as in the seminal experiments with photons at low energies, e.g. proving the violation of the famous Bell inequality, or of experiments in Particle Physics with entangled K-mesons (DAPHNE, CERN) or entangled B-meson (KEK) or entangled hyperons. So this brings up the question about figure 1, which states to show the simulation for three different cases compared to the experimental data:

1.) What is the difference between the blue and the red curve? In the supplementary material they claim to use a formula given in a reference 1 in Table 1 but never show this formula? This is a quite annoying procedure, having readers to look for a publication from the year 1957 (!) for a formula which is neither a lengthy one nor a correct one as the authors states in a footnote. There are also several formulae in Table 1, so it is not clear which they actually take and what the background of this formula is.

2.) I guess the green curve, thus represent a fully unpolarised photon pair, then the question arises, why there is a heap at about $+10^\circ$ which perfectly matches with the experimental data points, that also have here a heap?

3.) Why is there an asymmetry between negative and positive Delta phi? From the formula (2) one would not expect any.

C) Concerning Figure 2, the result when a scatter is inserted.

1.) Again there is quite an asymmetry between negative and positive Delta phi. Even stronger for the scatter data and for the positive region, not agreement with the asymmetry without scattering medium.

2.) Why is the error smaller for the non-scatter data, though the region of the polar angle has been broadened, it should increase!

3.) The error of the scatter data is very big compared to the non-scattered one. How does this come?

D) Concerning the result presented in Figure 6: Here the theoretical predictions are compared with their Geant Simulation, but not with the experimental data. Why? It seems that the red curve is half of the blue curve. Why should the case that the authors call "non-entangled" give half of the "entangled" case?

E) The authors seem also not to be aware of other papers dealing with entanglement in Compton interactions:

*) B.C. Hiesmayr, et al., "Witnessing Entanglement In Compton Scattering Processes Via Mutually Unbiased Bases", Scientific Reports 9, 8166 (2019): Here the authors claim that a separable scenario can reproduce the prediction of equation (2) exploited in the draft.

*) P. Caradonna, et al. "Probing entanglement in Compton interactions"; Here the authors also derive predictions in Compton scatterings of hypothetical separable, mixed and entangled states.

In summary, the paper claims a lot but the presentation and the experimental support is very poor, for that reason I do not suggest a publication if the authors cannot answer to all question in a satisfactory manner.

Reviewer #3:

Remarks to the Author:

This manuscript reports on the first experimental measurement, complemented with Monte Carlo radiation transport simulations, of quantum entanglement effects in Positron Emission Tomography (PET). The experimental results are original and, in my opinion, have the potential for high impact across multiple fields, including quantum physics and medical imaging physics. An important aspect of this work lies in measuring entanglement of 0.511 MeV gamma photons, which are at energies much higher than those of optical photons used extensively in quantum physics experiments. Indeed, annihilation quanta were the first to be identified for entanglement experiments and paved the way for subsequent experiments at optical wavelengths, which are now standard. The results may provide new directions for quantum physics experiments, and may eventually lead to the development of next-generation PET technologies that exploit entanglement to improve image quality for detecting diseases such as cancer and Alzheimer's. For these reasons, the manuscript warrants serious consideration for publication in Nature Communications.

Notwithstanding my encouraging remarks, several clarifications are required to bring the manuscript up to the expected standard:

1. Eqn. (2): Pryce & Ward (1947) originally described this as the [differential] double scattering cross section, to distinguish it from the more familiar single scattering cross-section. I suggest to clarify this eqn, either by using the same notation (σ_{double}) or similar (e.g. σ_{joint}). Similarly, in the sentence immediately above eqn (2), add "double" or "joint" before "cross-section", to avoid any confusion. Also, after eqn (2), last line on p3, I suggest the change: scattering cross-section  scattering probabilities

2. Fig. 1 - suggested improvements:

- (i) change vertical axis label to "normalised coincidence count rate" (as presumably this is what you're actually measuring)
- (ii) Geant4 data for all 3 curves need statistical uncertainties.

3. p.6, footnote c: while the study by McNamara et al (2014) did not consider entanglement, the follow-up study by Toghiani et al. (2016) did. Their theoretical calculations are consistent with Bohm & Aharanov (1957) and with the measurements of Snyder et al (1948), namely ratio of max/min count rates = 2.8 for theta near 90. In their Geant4 implementation, event statistics are only counted for joint/double scattering of entangled photons, as a means to effectively simulate the joint PDF for azimuthal angles. So that work should not be described as "incorrect". Ditto for footnote d on p10.

4. Figs. 2 & 6: statistical uncertainties in the QE-Geant4 data are needed

5. References: please cite this article: S. Sofer, E. Strizhevsky, A. Schori, K. Tamasaku, and S. Schwartz, Quantum Enhanced X-ray Detection, Phys. Rev. X 9, 031033 – Published 23 August 2019.

Minor comments/suggestions:

Figs 4 & 5: labels are missing and legends could be a little larger (especially (b)).

Fig. 6, caption typo: orthoganaly  orthogonally

Additional experimental studies to consider citing:

Wu and Shaknov, The angular correlation of scattered annihilation radiation (1950)
Kasday et al., Angular Correlation Compton- Scattered Annihilation Photons and Hidden Variables (1975);
Bertolini et al., Correlation of Annihilation γ -ray Polarization (1981)

Response to referee comments :

We thank the referees for their detailed review of the manuscript and answer their detailed points below. We have attached a revised manuscript accounting for the referees critique which includes new figures and analysis:

Reviewer #1

This paper describes experimental work and simulations that demonstrate the theoretically predicted angular correlation of photon polarization of annihilation gamma rays from positronium decay due to quantum entanglement. It is claimed, correctly to the best of my knowledge, that this is first time this has been demonstrated. This work is presented in the context of PET medical imaging and additional simulations indicate a potential application whereby the angular correlation can be used to distinguish true coincidence events from scattered and random events in PET. Despite the uncertainty as to whether or not this could one day become a practically useful technique, this is clearly a very exciting result and will be of great interest, possibly well beyond those working in the field(s) . There are various issues that should be addressed in manuscript. The manuscript appears technically sound, and the emphasis is clearly on the main result, however many of the descriptions on both results and context are quite brief and could be made clearer/enhanced by providing more technical information, also there are a few technical issues that should be addressed. These are listed below

1. line 6 - refs 2 and 3 are quite dated - please provide more up to date ones

These have been updated

2. line 14- radioisotope>radionuclide

Fixed

3. line 24 - ref 3 is only about scatter;

The referencing has been corrected to also include papers discussing random coincidences

4. As stated, scatter and randoms can make up a large fraction of the collected data in PET. There are established effective ways of correcting for the systematic errors they introduce, and the most significant detrimental they have is in reducing the final image SNR (characterized by 'noise equivalent count rate'). It is indicated how the additional information obtainable with this method allow true and scatter/randoms distributions to be determined, but as there does not seem to be the potential to discriminate on an event by event basis (only probabilities are obtained) and explicitly reject scatter/randoms, the effect on SNR will be limited. Rather than performing simple scaling/subtraction-type corrections, the additional information would now be exploited via a more sophisticated reconstruction procedure, for example incorporating the process into the forward model. I appreciate the proposed method is primarily for demonstration purposes, but these issues should be referred to in the discussion

We agree with the referees' comments and the next stage efforts to implement into more sophisticated imaging methodologies (e.g. maximum-likelihood expectation-maximization) is an important next step to optimise the use of the new entanglement information. We have added text to the revised manuscript to better clarify this point. Specifically we have commented how the imaging results in the paper were achieved with a realistic radiotracer activity and acquisition time as for a clinical PET scan (so that the results presented are a baseline indicator of the accuracy achievable in a realistic situation). We also added text to clarify that this first indicative study of imaging with QE-PET only uses a fraction of the data (from the selective θ range and utilising only two $|\Delta\Phi|$ bins), and outline how this would be optimised further for the next stage imaging programme with MLEM. Such further studies are planned but are a significant undertaking that is beyond the scope of this first paper (also see our response to referee 2 question B).

We should also remark that the backgrounds (scatter and random) are typically smoothly varying. Therefore they are often implemented into the more advanced imaging methodologies using a shape function (e.g. gaussian). Therefore the bin-by-bin extraction from QE-PET (as presented in the paper) will more likely be used to inform the magnitude and form of this background function rather than subtracting its contribution from the data on a bin-by-bin basis.

5. line 51- there has been a lot of recent discussion in the literature on this topic that is not referred to here (ie and should be) eg. Hiesmayr & Moskal Scientific Reports | (2019) 9:8166 | <https://doi.org/10.1038/s41598-019-44570-z>; Caradonna et al 2019 J. Phys. Commun. 3 105005

We agree that these recent references, which relate to the underlying QM theory, should be included and add further impact to the new work. We thank the referee for pointing this out and have included them in the revised draft.

6. line 75 - it is appreciated that this detector system is intended primarily as a demonstrator, however there are only limited systems that could measure the azimuthal angles in a PET configuration so it would be useful to provide some other relevant parameters of the CZT system - notably what is the temporal resolution, what is the angular (azimuthal) resolution for the scattered gamma direction , can 1st v 2nd hits be reliably discriminated.

We agree and have added text to clarify these points. For the data we use a simple algorithm to assign first and second hit information, done in a consistent procedure for the experimental data and simulation. This is now discussed in the revised manuscript.

Additionally most current generation PET scanners acquire time-of-flight information - the implications of this should be commented on. This could be in the discussion.

We have included comments on time-of-flight PET in the revised manuscript (footnote i)
The current work is complementary to these efforts: it could be used in addition to these methodologies or as an alternative where sufficiently precise timing information is not recorded/accessible (e.g. with emerging semiconductor technologies such as CZT currently deployed in SPECT but where achievable timing resolutions hinder the applicability of PET time-of-flight methods).

The next stage is to include entanglement effects in more sophisticated imaging frameworks (including time of flight when applicable), at which point the assessments indicated by the referee become feasible. We have studies underway assessing the quality of entanglement information accessible from detector systems where time-of-flight information could be effectively employed in imaging (e.g inorganic crystal arrays of LYSO detectors), but this is still in progress and beyond the scope of this first paper.

7. For readers unfamiliar with this topic a diagram indicating the detector setup and the various angles referred to would be useful

We agree and have included this

8. In the description on page 3, describe also the correlation expected if entanglement is not included, ie to provide context for the red line in fig 1. Also refer to the residual correlation after scattering

The analytic formula for non-entangled (but orthogonally polarised) γ pairs derived by Bohm and Aharonov has been added to the manuscript in the supplementary materials. The form of the $\Delta\Phi$ modulation is common to that of the entangled case, but the amplitude is significantly diminished. We note simulations employing the standard Geant4 classes are in good agreement with the analytic predictions for non-entangled photon pairs, as presented in the supplementary materials.

9. line 71- comment on the reduction in total true coincidence sensitivity resulting from the restriction on theta to around 82 degrees:

We have added this additional information to the revised manuscript. For illustrating the method and testing the entanglement modelling a narrow θ range was employed for this first analysis. The fraction of events where both photons scatter in the selected θ ranges (neglecting events which fired adjacent pixels) is around 6%. For a single detector the total fraction of hits which fire greater than one pixel is 50%.

In future analysis (e.g. maximum-likelihood expectation-maximization) a wider θ range can be employed, with a balance between event yield and the magnitude of the resulting $\Delta\Phi$ modulation for the event sample. We have added text and a reference to better indicate this.

Future work could also recover more yield from the neglected adjacent pixel hits, but is a significant programme of work requiring detailed modelling of the pixel hits (energy, position, timing and charge sharing).

We should add that the sensitivity is not compromised as QE-PET simply provides additional information to the standard (single-cluster) yield typically used in PET (also see response to Qn. 18)

10. Fig 1 - this figure is the key result of the paper and should better presented. In particular - what do the error bars signify;

The vertical error bars on the data points show the statistical error for the data in each bin. The horizontal bars show the bin width. We have now added error bands to also show the statistical accuracy of the simulated data. We have also improved the clarity of the figure.

how is the normalisation done exactly ;

We have added additional explanation to the caption to unambiguously state how the normalisation was carried out.

what is the measured 0-90 difference of the theoretical 2.85, and what is the error on that (if there is agreement, then ideally there should be statistical measure, the text just states 'good agreement') ?.

The QE-GEANT4 simulation that is compared to the data is based on the exact Pryce and Ward/Bohm and Aharonov predictions (as shown in the supplementary materials). Therefore, the agreement between the data and simulation provides a validation of this theory, from which the underlying (maximum) enhancement of 2.84 derives (reached for symmetric polar scatter angles of 82 degrees).

We observe a "measured" enhancement factor of 1.85 ± 0.04 (Fig 1). Note this is reduced compared to the maximum 2.85 value due to the θ range selected (not restricted solely to the maxima region around 82 degrees), the $\Delta\Phi$ resolution as well as the secondary effects from non-Compton processes, multiple scattering in the CZT, etc. As far as we are aware, this is the first test of this fundamental quantum mechanics prediction when all these effects are simulated.

In order to better quantify the agreement between the model and experimental data we have now included the overall χ^2 value in the text. Also, in the supplementary materials, we have added a plot showing the residuals between the data and model on a bin-by-bin basis.

What is the significance of this in the context of recent publications , eg Caradonna et al in 2019 (above) state, 'to date, no Compton scattering experiment of annihilation photons is known that has directly confirmed the predicted value of $R=2.85$ ':

The new data does provide some first confirmation, with the caveats as outlined in the response above. We plan future CZT high statistics measurements employing more limited θ scatter ranges to explore these fundamental issues and assess the maximum enhancement factor which can be measured with the apparatus. We feel such (planned) measurements are a longer term programme requiring extensive further data taking and are beyond the scope of this first paper. However, we share the referees' enthusiasm to achieve them.

The new apparatus has the benefit that a wide sample of θ range can be identified for both γ while previous measurements had very specific angle ranges and small acceptances. Clearly the planned high statistics data set has the potential to provide new levels of constraint on the basic quantum mechanics.

11. line 100 - a diagram would be helpful here:

We agree and have included this

12. line 107 - analysis cuts ARE SHOWN:

Fixed

13. fig 2 - Please provide some interpretation of what is seen here. Presumably the peaks in simulated data at +/- 90 degrees are assumed to be real ? ; move the caption so it doesn't cover the data.

We have now included the statistical error bands for the model predictions and moved the caption. With the achievable statistical accuracy of the single data point, we cannot make firm conclusions about a peak and its prominence shows sensitivity to the binning of the data. For the simulation we have better achievable statistical accuracy. The revised figure now uses a finer binning for the QE-GEANT4 prediction (possible as this is obtained with better statistics than the experimental data), so the detailed shape and magnitude of the prediction is clearer for the reader.

14. line 140 - restate here the theta acceptance criteria

Done

15. line 147 - I appreciate this is a simple demonstration of the potential of the technique , but please comment on the constancy of the scaling factors and how they would be affected by the residual correlated polarization in the scatter for different source and attenuator distributions

We have now included more quantitative discussion of these aspects in the supplementary materials. As discussed there we have studied the constancy of the residual polarisation through the phantom volume and find small variations. We also present an additional profile for a slice through a different region of the phantom using the same constant scaling factors as the results in the main paper.

As discussed in the paper and responses to other reviewer comments, we plan to include the entanglement information in iterative imaging in the future where attenuation etc can be studied in more detail than possible with these first results.

16. Fig 4 - Clarify the procedure here and in the text - the scaling factors refer to the fraction of T, S in the various phi windows in the raw data. They are not applied to the profiles in the reconstructed images shown in the figure (ie if that is the case, then it is incorrect)

We have expanded the text as suggested. The scaling factors refer to the relative amounts of T (or S) as a function of $\Delta\Phi$, not the amounts of T relative to S. This distinction has been made clearer in the main text and the description of the method of separating the true and scatter contributions has been expanded in the supplementary materials.

In the paper, we compared the extracted scatter profiles from the QE-PET method with the profiles from identified scatter events (accessible from Geant4 but not in real data). As presented in the paper good agreement is observed giving confidence in the method. Any residual differences in the profiles appear small compared with the statistical accuracy of the data. As discussed in previous responses the next stage will be to implement QE-PET in more advanced imaging methods.

17. Fig 4 C - This profile does not resemble what a scatter profile typically (ie always!) looks like (either through the image or through the raw data). In general the scatter profile will not reflect detail of the emission distribution (ie which 4c does). For example see the typical

profiles for a similar object/scanner geometry shown in fig 7 of Bentourkia et al *Computerized Medical Imaging and Graphics* 33 (2009) 477–488. Possibly this is a result of the energy spectrum of the detected scatter for this unusual detector set up, but it requires an explanation

We have included a revised figure in the new draft and thank the referee for prompting this investigation. The scatter events identified in the simulation did not discriminate between scatter before interaction in the CZT from those where a photon escapes the CZT and subsequently interacts in the phantom (comprising around 1% of the scatter yield). To be consistent with the previous works we have corrected this so that scatter only refers to events occurring prior to interaction in the detectors. The detail of the emission distribution is not present in the revised plots. The revised scatter profiles now display a purely Gaussian like distribution which is consistent with the relevant reference shared by the referee. The overall conclusions of the paper are unaffected.

The other artefacts (e.g. fluctuations in the profile) are typical for a scanner geometry in which there are gaps in acceptance and the detector planes do not form a perfect cylinder. The following reference providing a more complete discussion of such effects in FBP imaging was added to the paper:

Hallen P, Schug S, Schulz V. Comments on the NEMA NU 4-2008 Standard on Performance Measurement of Small Animal Positron Emission Tomographs. *EJNMMI Phys.* 2020; **7**.

So as not to convolve FBP post-processing effects with the quantum entanglement we applied minimal processing to the image data in the paper. There have been attempts to do post-processing of sinograms in a number of previous works (see refs below). It is expected that the QE-PET images reconstructed with the MLEM algorithm (future programme) will be a more optimum way to account for such acceptance artefacts.

Buchert R, Bohuslavizki KH, Mester J, Clausen M. Quality assurance in PET: Evaluation of the clinical relevance of detector defects. *J Nucl Med.* 1999;**40**:1657–65.

Edhoim PR, Lewitt RM, Lindholm B. Novel properties of the Fourier decomposition of the sinogram. *Proc Soc Photo Opt Instrum Eng.* 1986;**671**:8–18

Karp JS, Muehllehner G, Lewiti RM. Constrained Fourier space method for compensation of missing data in emission computed tomography. *IEEE Trans Med Imaging.* 1988;**7**:21–25.

18. line 194 - This is an exciting result but the discussion should also comment on the large price in sensitivity that is paid due to the various selection criteria applied to the data, in order to extract the angular correlation information. It would be useful to refer to Moskal et al *Eur. Phys. J. C* (2018) 78:970 <https://doi.org/10.1140/epjc/s10052-018-6461-1>, where these issues are discussed in detail .

We have included the suggested reference in the revised manuscript. We should remark that there will not be a price for overall sensitivity as the “standard” PET events used to form an

image (typically single pixel hits) will still be detected and analysed. The new entanglement information outlined in the paper is complementary to these events and provides independent information on the size and shape of the backgrounds for images derived from single pixel events as well as the QE-PET yield.

The simulations in the paper considered 1 trillion annihilations which we chose as it is comparable to a typical PET scan (a radiotracer activity of a few 100 MBq integrated over ~30 minute of acquisition). Therefore the accuracy of the scatter and random background distributions extracted in the paper should be indicative of what is achievable in a real scan, even when using the rather restrictive cuts employed in this first study. Future work using more advanced imaging methods could certainly better optimise the use of the new information, but is a significant future programme of work. We have added text to the revised manuscript to better clarify these points to the reader.

19. line 293 - CT scans aren't required/performed to determine scatter (well not additional ones anyway)

We have removed the word additional

Reviewer #2

A) The idea of using the inherent correlation of two gamma quanta to remove background is not new. It has been tackled by e.g. [Positron emission tomography coincidence detection with photon polarization correlation

Aimee L. McNamara, Kinwah Wua, David Boardman, Mark I. Reinhard and Zdenka

Kuncic:

<https://www.spiedigitallibrary.org/conference-proceedings-of-spie/8668/86681U/Positron-emission-tomography-coincidence-detection-with-photon-polarization-correlation/10.1117/12.2007794.short> or reference 16 cited in the draft.

There have been previous works exploring through simulation how the orthogonal polarisation correlation (non-entangled) between the gamma can provide benefits in PET imaging as pointed out by the referee. However, the simulations used in such works employed standard Geant4 classes. Such simulations would follow the 'non-entangled' prediction (red line/points) in Fig 6 (in original manuscript, now figure 8). We show in this work (Fig. 2) that these give a poor description of the measured double-Compton scattering probabilities, as they neglect entanglement. Such entanglement is explicit between the two-gamma from the positron annihilation processes. We feel our simulation developments represent a significant advance for developing quantum entangled PET, and a broader application in fundamental physics investigations of entangled photon-pairs at the MeV scale. They are the only entangled simulations in this regime, and we show that they are necessary to describe the experimental data. The analytical formula used in the previous references do however implicitly include the entanglement but it was not implemented in their simulations. IN the revised text we have updated the references to previous work to clarify this.

In addition, there are no quantitative statements regarding a possible improvement of scatter fraction reduction. There is also no discussion about the decrease of the PET sensitivity due

to the measurements of scattered photons and the reduction of events due to the limitation of the relative angle between scattering planes to a range close to 90 degrees.

Please see our response to reviewer 1 Q18 which addresses the same point.

This spatially resolved information on the random/scatter background (using the subset of data within the chosen θ cuts and realistic PET radiotracer cumulated activity) is in addition to the standard PET data. We show the principle of its extraction here - but expect a key use of this extra information in a commercial PET system would be in algorithms for iterative imaging such as MLEM. This next step is underway - but is a longer term programme beyond the scope of the current paper.

B) The authors claim that for "the first time" via Geant-4 the entanglement has been taken into account, which they denoted as QE-Geant4 (QE...quantum entangled). First, the programme is not published with the paper, they announce that it will be published (with their company?).

The adaptations we made to the Geant4 software (which included modifications to the underlying structure of the Geant4 code to allow communication between entangled particles) will be freely available to the community in a future public release. We expect this will become the standard for all future PET simulations - so it is important that our work is published to a wide audience. We achieved this with the consultancy of John Allison (a well-respected member of the Geant4 collaboration), so that the developments could be included in a more forward compatible manner within the Geant4 structure, but the developments will be open source.

Secondly, an MC can only simulate quantum mechanical effects (it is per se a classical computer programme) and is used in all standard experiments where entanglement plays a role such as in the seminal experiments with photons at low energies, e.g. proving the violation of the famous Bell inequality, or of experiments in Particle Physics with entangled K-mesons (DAPHNE, CERN) or entangled B-meson (KEK) or entangled hyperons.

Our work provides the first simulation of entangled photons at the MeV scale - we did not intend to claim that it is the first over all scales and are aware of the large community working with entangled photons in the optical regime. We have explicitly added photon entanglement and the energy scale when mentioned in the text.

The GEANT developers have confirmed that no other code for entanglement has ever been included in the simulation. The particle physics experiments regarding entangled particles may have used simulations to model detector acceptances - but without implicit simulation of the effects of quantum entanglement.

So this brings up the question about figure 1, which states to show the simulation for three different cases compared to the experimental data:

1.) What is the difference between the blue and the red curve? In the supplementary material they claim to use a formula given in a reference 1 in Table 1 but never show this formula? This is a quite annoying procedure, having readers to look for a publication from the year 1957 (!) for a formula which is neither a lengthy one nor a correct one as the authors states in a footnote. There are also several formulae in Table 1, so it is not clear which they actually take and what the background of this formula is.

The difference between these curves is the key witness of the entanglement effects. The blue curve shows the entangled expectation and the red curve gives the non-entangled (but oppositely polarised gamma) prediction. The red curve shows what one would obtain from the standard Geant4 code and clearly does not describe the data. We hope that this additional information also addresses the reviewers comment that such a simulation has been carried out before. All previous simulations using standard Geant4 would correspond to the red curve. We have included extra text in the revised manuscript to better clarify these points to the reader.

2.) I guess the green curve, thus represent a fully unpolarised photon pair, then the question arises, why there is a heap at about $+10^\circ$ which perfectly matches with the experimental data points, that also have here a heap?

This is a detector acceptance effect which is also evidenced in the experimental data. We carried out the unpolarised simulations so that any detector acceptance effects in the measured $\Delta\Phi$ distribution can be identified. We have added text to the paper to clarify these points and thank the reviewer for suggesting to include this.

3.) Why is there an asymmetry between negative and positive Delta phi? From the formula (2) one would not expect any.

Since submission we identified the source of this small asymmetry in the experimental data and obtained a new data set (used to derive the results in Fig 2 and 4 in the revised draft). We thank the reviewer for encouraging this investigation and we believe the new data even better highlights the data quality accessible with the new quantum technologies such as CZT.

C) Concerning Figure 2, the result when a scatter is inserted.

1.) Again there is quite an asymmetry between negative and positive Delta phi. Even stronger for the scatter data and for the positive region, not agreement with the asymmetry without scattering medium.

We do not expect agreement between the data with and without scattering medium as the latter would have some breaking of the entanglement. From the data it is indicated that when one of the entangled gamma scatters prior to detection the $\Delta\Phi$ enhancement is significantly reduced. To improve the clarity to the reader, we have modified the text and also added the QE-Geant4 prediction for the non-scattered data in the same (wider) polar angular cuts as the data.

The key result is that the scattered data is in agreement (within the achievable statistics) with the prediction from QE-GEANT4 in which implicit entanglement breaking is assumed at the first scatter (in the nylon). To the best of our knowledge this is the first experimental constraint on entanglement breaking of entangled photons at the MeV scale.

2.) Why is the error smaller for the non-scatter data, though the region of the polar angle has been broadened, it should increase!

If the polar angle bin is broadened then there will be a larger data sample in each $\Delta\Phi$ bin and the statistical errors would be expected to reduce rather than increase.

3.) The error of the scatter data is very big compared to the non-scattered one. How does this come?

This is due to the effect of the detector acceptance on the measured yield. We only accept events which have scattered in the nylon and then into the detector acceptance of the CZT. The solid angle for this significantly reduces the number of events. To give a sense of scale the data in Fig 2 (original manuscript, now figure 3) came from a month of running. We plan future programmes to increase the statistics, with new bespoke detector systems to measure scattered events. Although this is beyond the scope of the current paper, the current data does present the first constraint of photon pair entanglement breaking at the MeV scale. We feel this is an important result for quantum mechanics as well as for future application of entanglement in PET.

D) Concerning the result presented in Figure 6: Here the theoretical predictions are compared with their Geant Simulation, but not with the experimental data. Why? It seems that the red curve is half of the blue curve.

We only have measured experimental data within the detector acceptance and with the achievable detection characteristics of the CZT system. We therefore do not have experimental data to compare with these exact predictions as they would require a perfect detector with 100% efficiency and 100% accuracy in the hit positions.

Rather, Fig. 6 represents a verification of our modelling of the underlying entangled physics in the simulation. This is achieved by comparing the theoretical predictions (according to the entangled QM theory of Bohm and Aharonov) with the QE-GEANT4 simulation of a perfect detector. We have improved the text to make discussion of these results clearer for the reader.

Why should the case that the authors call "non-entangled" give half of the "entangled" case?

This difference is expected from the underlying quantum mechanics. We have expanded this discussion in the text and to further improve the clarity have also added the non-entangled theoretical formalism to the paper (see appendix).

E) The authors seem also not to be aware of other papers dealing with entanglement in Compton interactions:

*) B.C. Hiesmayr, et al., "Witnessing Entanglement In Compton Scattering Processes Via Mutually Unbiased Bases", Scientific Reports 9, 8166 (2019): Here the authors claim that a separable scenario can reproduce the prediction of equation (2) exploited in the draft.

*) P. Caradonna, et al. "Probing entanglement in Compton interactions"; Here the authors also derive predictions in Compton scatterings of hypothetical separable, mixed and entangled states.

We thank the referee for pointing this out and have included these references. These theoretical papers, in our opinion, add impetus to research of the type we present in the paper and indicate the high current interest in the community. Research of the type presented here, exploiting state-of-the-art detector systems, has the potential to address fundamental issues for entanglement in the future.

Reviewer #3 (Remarks to the Author):

This manuscript reports on the first experimental measurement, complemented with Monte Carlo radiation transport simulations, of quantum entanglement effects in Positron Emission Tomography (PET). The experimental results are original and, in my opinion, have the potential for high impact across multiple fields, including quantum physics and medical imaging physics. An important aspect of this work lies in measuring entanglement of 0.511 MeV gamma photons, which are at energies much higher than those of optical photons used extensively in quantum physics experiments. Indeed, annihilation quanta were the first to be identified for entanglement experiments and paved the way for subsequent experiments at optical wavelengths, which are now standard. The results may provide new directions for quantum physics experiments, and may eventually lead to the development of next-generation PET technologies that exploit entanglement to improve image quality for detecting diseases such as cancer and Alzheimer's.

For these reasons, the manuscript warrants serious consideration for publication in Nature Communications.

Notwithstanding my encouraging remarks, several clarifications are required to bring the manuscript up to the expected standard:

1. Eqn. (2): Pryce & Ward (1947) originally described this as the [differential] double scattering cross section, to distinguish it from the more familiar single scattering cross-section. I suggest to clarify this eqn, either by using the same notation (σ_{double}) or similar (e.g. σ_{joint}). Similarly, in the sentence immediately above eqn (2), add "double" or "joint" before "cross-section", to avoid any confusion. Also, after eqn (2), last line on p3, I suggest the change: scattering cross-section  scattering probabilities

Done

2. Fig. 1 - suggested improvements:

(i) change vertical axis label to "normalised coincidence count rate" (as presumably this is what you're actually measuring)

(ii) Geant4 data for all 3 curves need statistical uncertainties.

Done

3. p.6, footnote c: while the study by McNamara et al (2014) did not consider entanglement, the follow-up study by Toghyani et al. (2016) did. Their theoretical calculations are consistent with Bohm & Aharanov (1957) and with the measurements of Snyder et al (1948), namely ratio of max/min count rates = 2.8 for theta near 90. In their Geant4 implementation, event statistics are only counted for joint/double scattering of entangled photons, as a means to effectively simulate the joint PDF for azimuthal angles. So that work should not be described as "incorrect". Ditto for footnote d on p10.

In the revised text, we better highlight how this new work advances on this previous polarised (non-entangled) simulation study for Compton PET. The paper of McNamara et al. (2014) and Toghyani et al. (2016) both state that only standard Geant4 classes are used. As shown in Fig. 3 of the paper of Toghyani et al., the modulation of $\Delta\Phi$ is still there, but will be

diminished in magnitude compared to the proper entangled treatment. Their Geant4 simulation results would correspond to the non-entangled results (e.g Fig 2,6) of our current paper.

We agree that the theoretical discussions in the previous paper based on analytic estimates employed the full prediction including entanglement. We have removed the phrase “incorrect” as we agree this may be taken in a manner not intended. We have also improved the clarity of the text when referencing the paper

4. Figs. 2 & 6: statistical uncertainties in the QE-Geant4 data are needed

These have now been added to the figure in the form of error bands.

5. References: please cite this article: S. Sofer, E. Strizhevsky, A. Schori, K. Tamasaku, and S. Shwartz, Quantum Enhanced X-ray Detection, Phys. Rev. X 9, 031033 – Published 23 August 2019.

We have included this article. We thank the referee for highlighting this.

Minor comments/suggestions:

Figs 4 & 5: labels are missing and legends could be a little larger (especially (b)).

Legends (and axis labels) have been enlarged.

Fig. 6, caption typo: orthoganaly  orthogonally

Done

Additional experimental studies to consider citing:

Wu and Shakhnov, The angular correlation of scattered annihilation radiation (1950)

Kasday et al., Angular Correlation Compton- Scattered Annihilation Photons and Hidden Variables (1975);

Bertolini et al., Correlation of Annihilation γ -ray Polarization (1981)

Wu and Shakhnov are now cited in the introductory sections.

Reviewers' Comments:

Reviewer #1:

Remarks to the Author:

the manuscript has been extensively modified and has satisfactorily addressed all the points in my original review

Reviewer #2:

Remarks to the Author:

There are two big drawbacks to this contribution.

Firstly, the main result 'the simulation algorithms to detect two gammas by a CZT PET' is not made public (Authors promise to do it later) nor described well in the article.

Secondly, the Authors claim to prove the 'entanglement' which is not true.

Moreover, the Authors oversell their result by the notion 'quantum entanglement'. The real merit of the Authors' achievement is that the algorithm (if correct what I cannot verify from the information given in the article) can suppress in-patient scattering and to some extent random scatterings.

Moreover, it is not new to take into account 'this entanglement contribution' as quoted correctly in the appendix (line 422), but it is presented as new in the main part and in the reply.

Last but not least, the comparison with removing the part seemingly due to 'entanglement' in the algorithm is obviously no proof that entanglement is in the game. The authors simply ruled out two other possibilities.

- What does the term 'quantum entangled simulation' mean? Simulation is an algorithm running on a classical computer. Authors take formulae into account that are based on quantum considerations and modify the result of the GEANT framework?. Then compare it with the theoretical predictions, Fig.9.

- Figure 2 shows some inconsistency between ALL simulations in the range $D\phi=[0,180]$ and $D\phi=[-180,0]$ but not in the data. Fig.2. is claimed as the first verification of entanglement, this is certainly wrong since it does not detect entanglement, what Authors show is more precisely the consistence (up to the upper problem) with their algorithm.

- Introduction: Authors claim that the two gammas produced from positron annihilation carry SPATIAL entanglement. They go ahead and claim that via scattering the SPATIAL entanglement gets broken, but this is only a hypothesis.

- Experiment was performed with two detectors. The simulation algorithm took into account an arrangement of more of such detectors. Why the results of such simulations are called pre-clinical images? Preclinical would mean rather experimental studies with animals or at least with phantoms.

- Line 49: 'In this first study, we used the linear polarisation of the γ as the experimental observable sensitive to the entangled nature of the photons and implemented the entanglement...' Now Authors talk about POLARISATION entanglement?

- Authors do not explain why equation (1) is assumed, i.e. what are the physical considerations leading to (1). This is also relevant, because (1) is from theory only in one case of the different production methods of two gammas with high probability predicted, see historical references [13-15]. So the interpretation relies on the assumption that ^{22}Na source produces only entangled gammas.

· Authors claim to assume in the algorithm of GEANT 4 to collapse fully the state, the implications of this assumption should be investigated, which may change the whole interpretation.

· How is the (classical) communication within GEANT 4 framework changed such that this classical communication takes into account entanglement features correctly?

Reviewer #3:

Remarks to the Author:

Thank you for addressing all my comments, queries and suggested edits. The manuscript has improved considerably and I have no further comments or queries. I have a few more minor suggested edits below, which I think will improve readability and clarity. Otherwise I recommend publication.

Minor suggested edits:

- Abstract: through detailed simulations and experiments
- Intro: line 13 - high contrast  high sensitivity; line 17 - $e^+ + e^-$  e^+e^- ; line 23 - move ($\sim eV$) to between "optical" and "photons"; line 30 - contrast  signal-to-noise; line 43 - citation to refs. 10, 11 looks clumsy, so remove parentheses and "e.g."; line 57 - double differential cross-section describing the probability of Compton scattering.
- Fig. 1 caption: line 68 - the two entangled γ photons; line 72 - mutually perpendicular
- Results: line 77 - Monte Carlo simulation package  Monte Carlo radiation transport package; line 83 - e^+ annihilation  e^+e^- annihilation (ditto for line 217, Discussion; line 84 - Kromek  Kromek Group (ditto for Competing Interests).
- Conclusion: line 248 - This paper presents a first simulation of  This study presents the first results on

Zdenka Kuncic

Response to individual Reviewer comments

We respond to the reviewer's comments below. The reviewer comments are shown in red text and changes to the manuscript are shown in blue text.

Reviewer 1

the manuscript has been extensively modified and has satisfactorily addressed all the points in my original review

We thank the reviewer for the useful comments on the manuscript and we are happy they are satisfied with the revised version.

Reviewer 2

We respond to all of reviewer #2's comments below (we collate related comments to avoid unnecessary repetition in our answers). We note that a significant number of new questions were posed in this final round.

What does the term 'quantum entangled simulation' mean? Simulation is an algorithm running on a classical computer. Authors take formulae into account that are based on quantum considerations and modify the result of the GEANT framework?. Then compare it with the theoretical predictions, Fig.9. How is the (classical) communication within GEANT 4 framework changed such that this classical communication takes into account entanglement features correctly?

Yes - "Quantum entangled simulation" means that the simulations take into account the predicted effect of the quantum entanglement between the two annihilation-gamma in their first Compton scatter process. We state this in the paper (L75), quote the theoretical entangled cross section we implemented in Geant4 (equation 2) and the reference where it is derived. As a verification of these developments, we show (Fig. 9) that the simulation predictions are entirely consistent with the analytic theoretical formulae for double Compton scattering derived by Bohm and Aharonov (equation 2). As a result, we are confident the entanglement is implemented correctly in the simulation. It is perfectly valid to implement the predictions of entanglement effects for reaction processes into a "classical" computer simulation.

We note the description of the implementation of entanglement in the simulation was acceptable to Reviewers #1 and #3. However, we have changed the phrasing in the sentence below to perhaps improve the clarity for Reviewer #2:

Line 81 and reproduce -> to reproduce

Firstly, the main result 'the simulation algorithms to detect two gammas by a CZT PET' is not made public (Authors promise to do it later) nor described well in the article.

We shared the relevant source code of our new version of Geant4 in the GitHub repository (made available in the link provided in the email <https://github.com/jrbrown81/Quantum-Entangled-PET>) so it was available for reviewers. It would also be available to readers on request. As stated L. 79 and in the "Code and data availability" section, our developments will be made open source to the world community in the first Geant4 public release after publication. We don't really know what more we could do in this regard

However, we should remark that this code is just doing what we stated clearly in the text. In Fig 9 we show the consistency of the simulation output with *analytic* results from the entangled theory of Bohm and Aharonov (equation 2) - this clearly illustrates that the implementation in the simulation is consistent with established entanglement theory.

Figure 2 shows some inconsistency between ALL simulations in the range $D\phi=[0,180]$ and $D\phi=[-180,0]$ but not in the data.

The phrasing of the question leaves it open to a number of interpretations. We assume that the “inconsistency” reviewer #2 is referring to is not a continuation of the comment in a previous round questioning why the (entangled, non-entangled and unpolarised) simulations are different at all – which we answered in the previous round and is a consequence of the established entanglement theory.

If we take it that reviewer #2 is referring to some differences in acceptance effects between the 3 simulations in the different phi regions - then this is not unexpected. The unpolarised simulations have no correlation in the acceptance between the 2 detector heads. In contrast the polarised and entangled simulations have a *correlated* detector acceptance (i.e. the scatter angle in one detector influences the probability for the scatter direction of the other detector). Small differences in acceptance are therefore expected - and are modelled correctly in the simulation. We also note other effects may produce small differences in the $\Delta\phi$ acceptance (e.g. the detector crystals are mounted in different relative positions to the holding frame in each head (fig 5), mounting table only under the detector array, etc.). As the setup is accurately modelled in the simulation, such effects will be accounted for. The underlying theory is of course perfectly symmetric in $\Delta\phi$, but acceptance may not be perfectly symmetric. That is one of the reasons why our development of a simulation including entanglement effects is so important.

The comment “but not in the data” is also difficult to understand. The main result of Fig. 2 is that the new data is statistically consistent with the Bohm and Aharonov entangled theory (whose predictions were recently re-confirmed in Caradonna [11]). The comparison gives a $\chi^2 = 1.84$ (i.e. close to 1) and the plot of the residuals (Fig. 10) does not reveal significant $\Delta\phi$ dependent discrepancies within the statistical accuracy of the data (~11 of the 18 points have error bars which are consistent with zero). We could implicitly state that data and simulation agree within the achievable statistical accuracy of the data - but we would have assumed this to be obvious from the statistical analysis presented (Reviewers #1 and #3 had no issue). We do not discuss any differences between the experimental data and entangled predictions in the paper, as they are consistent within the statistics.

We don't see a benefit to the reader in presenting detailed statistical analysis comparing the data to the non-entangled and unpolarised simulation - clearly neither of these are able to reproduce the data.

It is not new to take into account 'this entanglement contribution' as quoted correctly in the appendix (line 422), but it is presented as new in the main part and in the reply.

We state clearly that we implement the established entanglement theory of Bohm and Aharonov in our simulation, which is a development of (and entirely consistent with) the earlier theoretical works [13,14] referenced on L422. We do not state it is the first time “this entanglement contribution” has been taken into account anywhere in the paper.

It is common in the literature to describe the entangled theory as that of Bohm and Aharonov (e.g. also see Ref 11), even though much of the formalism was developed earlier (e.g. Pryce and Ward [13]). We are not sure if reviewer #2 somehow has the impression that we present our own “new” theory for entanglement - this is not the case.

Fig.2. is claimed as the first verification of entanglement, this is certainly wrong since it does not detect entanglement, what Authors show is more precisely the consistence (up to the upper problem) with their algorithm.

We also do not state it is the “first verification of entanglement”. To address reviewer #2’s comments we have brought the discussion of previous measurements of the $\Delta\phi$ enhancement earlier in the text (L64).

“Previous measurements of this ratio in restricted kinematics of θ_{1} , θ_{2} and $\Delta\phi$ \cite{Wu1950,Bertolini1981} (and others summarised in ref 11) showed statistical agreement with the $\Delta\phi$ enhancement predicted by entangled theory, when calculated corrections for the detector efficiencies and backgrounds were included in the theory.”

These previous measurements gave statistical agreement with the predicted $\Delta\phi$ enhancement from entangled theory in more restricted kinematics (after including analytic corrections for acceptance and backgrounds). The main focus of these earlier works was to test the theory through measurement of the $\Delta\phi$ enhancement in specific kinematics, typically where the enhancement magnitude was maximised. The new CZT results (Fig 2) measure $\Delta\phi$ with unprecedented detector acceptance and compare with a new, detailed simulation including entanglement. In case reviewer #2 arrived at this misconception through comments in the draft “press release” for a non-physics audience we have made this text more specific in the revised manuscript.

We do not understand the phrases “detect entanglement” or “more precisely the consistence (up to the upper problem) with their algorithm”. “Detecting entanglement” is an oxymoron as doing so would destroy the entanglement. We can only infer entanglement from its predicted effect on measured observables using established theory, as done in the paper. The “algorithm” is in fact the established entanglement theory of Bohm and Aharonov.

The Authors oversell their result by the notion ‘quantum entanglement’.

Secondly, the Authors claim to prove the ‘entanglement’ which is not true.

Last but not least, the comparison with removing the part seemingly due to ‘entanglement’ in the algorithm is obviously no proof that entanglement is in the game. The authors simply ruled out two other possibilities.

It would be odd to ignore the statistical agreement between data and established entanglement theory observed in Fig. 2 (and the previous $\Delta\phi$ measurements in more limited kinematics), this is the very basis of the scientific method. Our new data show clear disagreement with the non-entangled and unpolarised predictions (as appreciated by the reviewer) – and clear statistical agreement with the predictions of the established entangled theory. The residuals from the comparison are clearly presented in Fig 10 - so we detail the level of agreement in full detail to the reader.

There are currently no accepted “alternative” theories of relevance to the entangled $\Delta\phi$ correlation tested and exploited in the paper. The paper is not aimed as an exploration of alternative or speculative theories. In case the reviewer is influenced by the refuted results of Ref 10 (Hiesmayer 2019), which discussed hypothetical non-entangled contributions, we have added the following footnote to the manuscript (p4 bottom) for clarification to the reader.

We note recent theoretical work \cite{Caradonna2019} explored additional hypothetical non-entangled (separable) contributions in detail, and obtained $\Delta\phi$ ratios consistent with Bohm and Aharonov (equation 2). The results of Ref \cite{Caradonna2019} and Bohm and Aharonov are in formal disagreement with the recent controversial conclusions regarding the role of non-entangled mixed

states for the gamma produced in positron annihilation (Ref \cite{Hiesmayr2019}). We therefore take the Bohm and Aharonov formalism as appropriate for this work.

As a further check in Figs 1-2 (below) we compare the predictions of Caradonna and Bohm and Aharonov. There are no discernible differences in the predicted $\Delta\phi$, consistent with the conclusions of Ref 11. Additional refutation of ref 10 is indicated by the unpolarised simulation results in Fig 2. In isolation the hypothetical (non-entangled) mixed state of Ref 10 corresponds to gamma with equal contributions from different polarisation states. This would be expected to have properties similar to the “unpolarised” gamma simulation presented in the paper, which also has random polarisation. This unpolarised simulation shows an intrinsic enhancement (R value) of 1, i.e. flat, in agreement with established quantum theory (Ref 11 and Bohm and Aharonov). It is also clear from Fig 2 that when the unpolarised simulations are passed through the CZT detector acceptance any “induced” $\Delta\phi$ structures evident in the measured data are far smaller than the $\Delta\phi$ modulations in the data arising from entanglement.

Introduction: Authors claim that the two gammas produced from positron annihilation carry SPATIAL entanglement. They go ahead and claim that via scattering the SPATIAL entanglement gets broken, but this is only a hypothesis. Line 49: ‘In this first study, we used the linear polarisation of the γ as the experimental observable sensitive to the entangled nature of the photons and implemented the entanglement...’ Now Authors talk about POLARISATION entanglement?

We don’t claim anything about spatial entanglement or even mention the phrase in the paper. Nothing we measure gives sensitivity to the spatial entanglement. Although spatial entanglement is implicit alongside the entanglement of polarisation, obtaining sensitivity to it in measurements would require exquisite coincidence timing resolution for the gamma detections. These points were not raised by reviewers #1 and #3, we therefore feel the comments reflect reviewer #2’s misunderstanding rather than suggesting a need to modify the text. The only possibility we can infer is that the use of the phrase “action at a distance” in the abstract was misinterpreted by Reviewer #2 as only applying to spatial entanglement and that he/she is unaware that such effects also have relevance to polarisation entanglement. Much of the world programme carrying out fundamental research into action-at-a-distance effects in the optical regime utilise entanglement of polarisation (e.g. see Jen Wie-Pan et. al.; *Science*, DOI: 10.1126/science.aan3211 and many references therein).

The real merit of the Authors’ achievement is that the algorithm (if correct what I cannot verify from the information given in the article) can suppress in-patient scattering and to some extent random scatterings.

We reiterate that “the algorithm” referred to by the reviewer is in fact the established entanglement theory of Bohm & Aharonov (recently re-confirmed in Ref [11]), we show clearly in the paper that the simulation implements it correctly (Fig 9), placed the code on a github repository and will release it to the community in the next GEANT4 release.

Our work presented in the paper does not “suppress” in-patient scattering - it shows a way to quantify its contribution to PET images in a spatially resolved way using only the collected data and inputs from the new entangled simulation.

We do not understand what the reviewer means by “random scatterings”. The random contributions discussed in the paper arise from uncorrelated gamma to the image (i.e. arising from different annihilation events). The reviewer seems to convolute these with scatter backgrounds.

Authors do not explain why equation (1) is assumed, i.e. what are the physical considerations leading to (1). This is also relevant, because (1) is from theory only in one case of the different production

methods of two gammas with high probability predicted, see historical references [13-15]. So the interpretation relies on the assumption that ^{22}Na source produces only entangled gammas

If by ‘assumption that ^{22}Na source produces only entangled gammas’ reviewer #2 is alluding to uncorrelated gamma coincidences entering into the data in Fig 2 (i.e. two gamma from different positron annihilations) then this is not relevant. The event rate in the detectors for the setup acquiring the data of Fig 2 is around 20Hz. The contribution of such random (non-entangled) coincidences from the source is therefore at the 0.2% level and has no impact to our results or conclusions.

We wondered if the reviewer may be concerned about a contribution from the recently speculated (and immediately refuted) contribution of hypothetical non-entangled mixed states (Ref 10). In which case we should reiterate that the Bohm and Aharonov formalism gives consistent results with the more recent Ref. 11 which subsequently explored such hypothetical non-entangled contributions (and other second order effects) in positron annihilation in detail. In Ref 11 it is shown that including such hypothetical contributions does not produce any contradiction with the Bohm and Aharonov formalism ¹. The footnote on p4 clarifies the situation regarding the refuted Ref 10 to the reader. As mentioned above, for the avoidance of doubt, we have coded the Caradonna theory [11] (including non-entangled contributions) and compare with Bohm and Aharonov (Figs 1-2 below). There are no discernible differences with the Bohm and Aharonov theory we adopt in the paper.

Regarding the 3 references mentioned by the reviewer there is nothing contradicting the (subsequent) Bohm and Aharonov entangled theory of positron annihilation in these papers:

Ref [13] Gives a prediction of the cross section for double Compton scattering which is identical to Bohm and Aharonov (equation 2)

Ref [14] Presents theoretical expressions compatible with equation 2 in the paper and suggests experiments to be performed

Ref [15] is a 1929 paper by Klein and Nishina on polarised Compton scattering - used in the later works of Pryce and Ward, Bohm and Aharonov.

Experiment was performed with two detectors. The simulation algorithm took into account an arrangement of more of such detectors. Why the results of such simulations are called pre-clinical images? Preclinical would mean rather experimental studies with animals or at least with phantoms

Because we simulated a preclinical scanner geometry and a preclinical NEMA-NU4 phantom. The phantom is the industry standard for assessing the performance of preclinical systems. It is clear that these are simulated studies of a hypothetical preclinical system. Previous papers employing simulation use the same phraseology. Reviewers #1 and #3 had no issue with the description and we do not feel modifying the text is appropriate.

Authors claim to assume in the algorithm of GEANT 4 to collapses fully the state, the implications of this assumption should be investigated, which may change the whole interpretation

We clearly stated in the paper how entanglement breaking from a scatter process was implemented in the GEANT4 model and also provided the world’s first experimental data on such a process to test the predictions. The predictions for $\Delta\phi$ following a scatter from an initially entangled pair are consistent with the experimental data within their errors (Fig. 4), so we already provide a first investigation. We clearly observe a diminished amplitude of the $\Delta\phi$ distribution in statistical agreement with the simulation.

¹ We note that Caradonna *et al.* [11] highlight the potential existence of very small lobe-like structures for certain kinematic regions (e.g in Fig 4 in [11] showing the 4-dimensional cross section for Compton scattering of annihilation gamma). These are additional features arising from entanglement (i.e. not due to a hypothetical mixed state). They are also predicted in the Bohm and Aharonov theory and are therefore included in our simulation. However, their predicted effects are well below the sensitivity of the current data.

If it is somehow found that the entangled state is not fully collapsed by the scattering of a gamma, (despite the significant change in energy, direction and polarisation) then it may influence the (small amplitude) $\Delta\phi$ distribution for scatter events. However, this could naturally be accounted for in our scheme by repeating the scaling factor calculations using the updated scatter model. The “whole interpretation” or the method as presented in the paper would not change. The results relating to extraction of the random contribution to the PET image are independent of this small residual $\Delta\phi$ distribution, so would be unaffected. We have added a footnote to clarify this (footnote I P33).

We should remark that the size of this residual enhancement is modelled assuming the expected collapse of the entangled state, as supported by the results presented in Fig 4. If future developments in our understanding of entanglement breaking at the MeV scale result in changes to this small residual $\Delta\phi$ dependence then the method would be unaffected, but new scaling factors would apply (determined using a simulation containing the updated models of entanglement breaking).

Reviewer 3

Thank you for addressing all my comments, queries and suggested edits. The manuscript has improved considerably and I have no further comments or queries. I have a few more minor suggested edits below, which I think will improve readability and clarity. Otherwise I recommend publication.

Minor suggested edits:

- Abstract through detailed simulations and experiments Done
- Intro: line 13 - high contrast  high sensitivity; (we would prefer to keep the text as is because a key advantage of PET imaging is contrast and it has low sensitivity)
- line 17 - $e^+ + e^-$  e^+e^- ; line 23 - move ($\sim eV$) to between "optical" and "photons"; Done
- line 30 - contrast  signal-to-noise Done;
- line 43 - citation to refs. 10, 11 looks clumsy, so remove parentheses and "e.g."; Done
- line 57 - double differential cross-section describing the probability of Compton scattering. Done
- Fig. 1 caption: line 68 - the two entangled γ photons; line 72 - mutually perpendicular Done
- Results: line 77 - Monte Carlo simulation package  Monte Carlo radiation transport package; Done (but referred to it as radiation and particle transport package)
- line 83 - e^+ annihilation  e^+e^- annihilation (ditto for line 217, Done
- Discussion; line 84 - Kromek  Kromek Group (ditto for Competing Interests). Done
- Conclusion: line 248 - This paper presents a first simulation of  This study presents the first results on Done

We would like to thank reviewer #3 for the very useful comments - which have improved the paper.

Fig 1: The predicted enhancement factors (see paper for definition) as a function of scatter angle. The Caradonna prediction [11] is shown by the (red points) and Bohm and Aharanov (using the formalism in the Pryce and Ward paper [13]) by the blue points.

Fig 2: (left column) The predicted $\Delta\phi$ distributions from Caradonna theory (Ref 11). (right column) the Bohm and Aharanov entangled theory employed in the current paper (using formalism as presented in Pryce and Ward [13]). The first two rows show comparisons for two theta scatter angles relevant to the current work (i.e. within the theta scatter range used in the analysis). The lower row shows the data from Fig 1 without the overlay.

Reviewers' Comments:

Reviewer #1:

None

Reviewer #2:

None

Reviewer #4:

Remarks to the Author:

Dear editor and authors,

First of all, I would like to clarify that I am not an expert in PET imaging but I am in quantum information science. In the following lines, I will challenge the hypothesis of this manuscript from a physical and quantum information point of view. I will focus on the mathematical formalism used and the explanations that the authors give to motivate their research and results.

While reading this manuscript and after checking some of the details and the references provided, my feeling is that the authors are not familiarized with quantum information formalism and notions. For instance, in the introduction one can find statements of the form:

"The two annihilation are quantum-entangled, implying that a measurement of an observable for one of them instantaneously affects the properties of the other."

This is not how quantum entanglement works. This is a misleading description often used in popular science material. The collapse of one of the entangled parties does not affect instantaneously anything since it will violate causality. You can maybe interpret that philosophically, but physically, the use of entanglement as a resource requires a classical communication channel.

The central claim of this work is stated in the following sentence:

"In this work, we carry out a first simulation of quantum entanglement for photons in the MeV regime and explore the benefits of using this quantum information in analysis of PET data."

I can easily simulate two entangled photons with energies of MeV, so the first part of this claim is not novel. There exist several works that study fundamental interactions such as $e+e-$ or Compton scattering from a quantum information point of view. Some of them are already cited. Other related references are Cervera-Lierta et. al. *SciPost Phys.* 3, 036 (2017) (where the polarization amplitudes of all QED processes, including Compton scattering and $e+e-$ are presented at tree-level and the entanglement properties analyzed), Beane et al *Phys. Rev. Lett.* 122, 102001 (where entanglement suppression is analyzed in strong interactions) and Araujo et. al. *Phys. Rev. D* 100, 105018 (where entangled cross sections are analyzed).

Right after these claims, the authors write what I believe is one of the most controversial points of the manuscript:

"The useful events collected to form a PET image are the true (quantum-entangled) pairs having a LOR which crosses the annihilation site."

How can two photons (entangled or not in origin) hold the entanglement properties through a human body and be detected cm away? This is physically impossible, and the photons that arrive at the detector are surely not entangled. Their wave function collapses almost instantaneously after the pair is generated due to the noise and decoherence.

At this point, I assume that the only thing that can motivate this study is whether or not a PET image can distinguish those photons generated by the $e+e-$ to 2photons interaction from other photons coming from other processes or the background.

Starting with the e^+e^- to 2 photons reaction. This interaction can generate photons entangled or not. Depending on the angle between the photons, one can guess if their state was entangled or not in origin. Then, a reasonable approach might be to assume that the photons detected in a particular angle where originated from this process. However, this reasoning has its flaws. The first one is how can you distinguish the photons that come from this interaction from the background photons, being the only difference that some were originally entangled and not the others. As stated above, the photons that arrive at the detector are not entangled, so all photons detected are indistinguishable in terms of entanglement. Secondly, you will need also to discard (or quantify) any other possible physical process that can produce entangled photons and that can occur between the target and the detectors. Since a PET is not performed under isolated and controlled conditions, one should expect, in my opinion, other sources of entangled photons.

From this point, the authors misinterpreted the results of the references [1], [12,13]. First, they state "The annihilation [photon] have orthogonal linear polarisation with an entangled wave function which can be expressed as" and write eq(1) which corresponds with the single state of the two-photon linear polarizations. As I said in the previous paragraphs and as it is described in (for instance) Ref. [12], this interaction generates a pair of photons which entanglement properties depend on the relative angle between them. That is, these two photons can outcome as a product state of their polarizations or as an entangled state. Therefore, it is incorrect to assume that all outgoing photons of the reaction e^+e^- to 2 photons are entangled because they are not. I believe that the authors may have misinterpreted this result after reading reference [1], where only the maximal entangled case is discussed.

Another important point is which formula the authors are introducing to their simulations. The step from eq(1) to eq(2) is a mystery for me. It is not detailed how the authors use eq(1) to arrive at eq(2), which seems the central formula of all their analysis. The reference that they provide (Ref.[13]) is a less than a page article from 1947 with only one equation and without describing any derivation. That reference states "If the two gamma-quanta emitted on the annihilation of a slowly moving positron-electron pair both undergo a Compton scattering..." and I believe that the formula provided is another form of the Klein-Nishina formula. However, this formula applies to Compton scattering and has anything to do with the possible previous entanglement between the photons involved. The authors have incorrectly assumed that somehow eq(1) (maximally entangled photons originated in one process) is included in a cross-section of a different process.

At this point, giving the fact that the authors always talk about the "quantum-entangled photons" but not all photons from this kind of reaction are entangled, it is not clear how they can interpret the results under the initial claim "we carry out a first simulation of quantum entanglement for photons in the MeV regime". They are also mixing the formalism of two different processes, e^+e^- and Compton scattering.

The next paragraphs describe the details of the PET simulation. I am not familiarized with this kind of simulations, so I cannot give an educated analysis of it. However, my concerns stand: are the authors assuming that all photons generated are entangled? Does eq.(2) quantify properly the entanglement of these photons?

The authors also mention multiple times the "entanglement effects". What are these "effects"?

Coming back to the simulation and the results shown in Fig.(2) I have a question motivated by my ignorance about this kind of simulations, and two observations. The question is: is the data real data or simulated data (black points)? The first observation is that the authors distinguish two models, the entangled and the standard, but as I argue above, photons coming from e^+e^- interaction can emerge both entangled or not, so it has no physical sense to try to analyze them separately with two different models.

The notion of "entanglement breaking" has no physical meaning. Entanglement is not a property that can be broken but a property that can be lost due to other interaction processes. This is maybe a terminology observation unless the authors have another meaning for that.

"Additionally, a first measurement of photon entanglement breaking for photon energies above the

optical regime has been achieved". This manuscript does not describe in any place an entanglement measurement protocol. Moreover, photons do not arrive entangled to the detector, so talking about entanglement detection has no physical sense here.

Besides all the controversial points described above, I would like to add another one. To my understanding, the authors aimed to prove that entangled photons play a relevant role in PET imaging and that the use of entanglement formalism may prove useful in the reconstruction of these images. However, they do not discuss nor mention the time evolution of the hypothetical photon's wavefunction. This is not a particularly problematic point if all the analysis is performed at 0 time, that is, at the generation point (formalism already developed in some references, for instance, Ref. [10]), but the goal is to use it in a real-world application, where noise and decoherence are unavoidable.

For all the reasons stated above, I conclude that this work is ill-defined (uses well-known formalism incorrectly, for instance, by mixing Compton scattering cross-sections with entangled wave functions from a different process without any intermediate justification), the hypothesis is physically incorrect (photons that arrive at the detector are not entangled, so an entanglement analysis there has no physical sense) and, therefore, the conclusions cannot be interpreted under the hypothesis assumptions. I will not discard that a proper analysis of entanglement in processes like e^+e^- and its correct insertion in the PET simulations, might be scientifically relevant, but I am afraid that this work does not present the necessary formalism.

Reviewers' comments:

Reviewer #4 (Remarks to the Author):

Dear editor and authors,

1) First of all, I would like to clarify that I am not an expert in PET imaging but I am in quantum information science. In the following lines, I will challenge the hypothesis of this manuscript from a physical and quantum information point of view. I will focus on the mathematical formalism used and the explanations that the authors give to motivate their research and results.

While reading this manuscript and after checking some of the details and the references provided, my feeling is that the authors are not familiarized with quantum information formalism and notions. For instance, in the introduction one can find statements of the form:

“The two annihilation are quantum-entangled, implying that a measurement of an observable for one of them instantaneously affects the properties of the other.”

This is not how quantum entanglement works. This is a misleading description often used in popular science material. The collapse of one of the entangled parties does not affect instantaneously anything since it will violate causality. You can maybe interpret that philosophically, but physically, the use of entanglement as a resource requires a classical communication channel.

The instantaneous collapse of a spatially extended entangled wavefunctions are not only discussed in popular science articles. From the original proposition its implications are studied in many papers to the current day e.g. Richardson et. al. *Quantum Studies: Mathematics and Foundations* **1**, 57 (2014); Aspect, *Nature* vol 446, pages ~~866–867~~(2007), and many others. ~~Indeed, there are open philosophical questions,~~ and the role given to collapse (and even the wavefunction itself) depends on different interpretations (e.g. bare formalisms, the quantum Bayesianism adopted in many information theories, many-worlds theorems etc).

We are happy to remove any references to instantaneous if the referee takes objection to it – but he/she was the only reviewer who did. We note that whether collapse is found to be instantaneous or not has no consequence to the results presented. Also, we remark that instantaneous wavefunction collapse would not violate causality or it would already be disregarded. Information from the entanglement cannot be transferred faster than the speed of light.

2) The central claim of this work is stated in the following sentence:

“In this work, we carry out a first simulation of quantum entanglement for photons in the MeV regime and explore the benefits of using this quantum information in analysis of PET data.”

I can easily simulate two entangled photons with energies of MeV, so the first part of this claim is not novel. There exist several works that study fundamental interactions such as e^+e^- or Compton scattering from a quantum information point of view. Some of them are already cited. Other related references are Cervera-Lierta et. al. *SciPost Phys.* **3**, 036 (2017) (where the polarization amplitudes of all QED processes, including Compton scattering and e^+e^- are presented at tree-level and the entanglement properties analyzed), Beane et al *Phys. Rev. Lett.* **122**, 102001 (where entanglement suppression is analyzed in strong interactions) and Araujo et. al. *Phys. Rev. D* **100**, 105018 (where entangled cross sections are analyzed).

Simulating the entangled photons in isolation is not the main part of the simulation. The novel part is modelling the interactions of entangled photons with matter, and accounting for the predicted effects of entanglement in the relevant particle interactions. A simulation is different to papers on underlying theory as quoted by the referee. The fact the reviewer can “easily simulate it” is irrelevant - and rather doubtful given the knowledge of gamma interactions indicated by Q3.

3) Right after these claims, the authors write what I believe is one of the most controversial points of the manuscript:

“The useful events collected to form a PET image are the true (quantum-entangled) pairs having a LOR which crosses the annihilation site.”

How can two photons (entangled or not in origin) hold the entanglement properties through a human body and be detected cm away? This is physically impossible, and the photons that arrive at the detector are surely not entangled. Their wave function collapses almost instantaneously after the pair is generated due to the noise and decoherence.

The referee appears unaware of the differences between this energy regime and that of optical photons (for which the statement may be valid). For MeV scale gamma the mean free path between interactions is far larger than in the optical regime and the first interaction of the photon is likely to be at the detector. It is not “physically impossible”. The GEANT4 model includes all known processes at this energy scale (photoelectric, pair production, Rayleigh,..) so their contribution is quantifiable, small (and modelled) in the simulation.

We assume the referee is guided by studies in the optical regime, where of course such decoherence effects would indeed be dominant (the human body is opaque at these wavelengths with a mean free path much smaller than the size of the body). The statements of the reviewer are entirely at odds with the nature of gamma interactions.

The statement that the “wave function collapses immediately” is also incorrect. If this was the case (and certainly this is not proposed in any accepted literature we are aware of) then the $\Delta(\phi)$ correlations observed in the current work (e.g fig 2) and all previous measurements would be unexplainable by quantum mechanics. A collapsed wavefunction would give expectation of the non-entangled results in fig 2. This does not describe the current or any previous experimental data. The reviewer is just incorrect in his/her assertions about immediate wavefunction collapse.

4) At this point, I assume that the only thing that can motivate this study is whether or not a PET image can distinguish those photons generated by the e^+e^- to 2 photons interaction from other photons coming from other processes or the background.

We do not identify or distinguish individual photons from different sources. We use the $\Delta(\phi)$ correlations induced by the entanglement (and the differences in correlations to those from backgrounds) to separate the contributions on a statistical basis. This is clearly presented in the paper.

5) Starting with the e^+e^- to 2 photons reaction. This interaction can generate photons entangled or not.

There is (at the very least) one fatal flaw in the referee’s assertion that the e^+e^- into two photons can generate non-entangled photons; namely it violates parity conservation in electromagnetic processes. [e.g. Yang Phys Rev 77 242 1950] . The statement goes against the established theoretical work in the field which has been tested for many decades.

In principle there are indeed 4 entangled Bell states. However, only two of these are cross polarised (conserve angular momentum) and are therefore possible from the positronium annihilation. Parity conservation in the annihilation process dictates that only one of these contributes, which is the wavefunction combination shown in equation 1 (the other state with positive sign being forbidden). All possible Bell states were considered in the development of the theory we employed [e.g. **D. Bohm and Y. Aharonov Phys. Rev. 108, 1070, 1957**].

The referee is therefore incorrect in the assertion. Since the first publication of the Bohm and Aharonov paper in 1957 there is only paper contradicting the fully entangled nature of the photons from positron annihilation [Hiesmayr et. al., *Sci Rep* **9**, 8166 (2019)] which as part of the work presented speculated about a mixed state (which was actually considered and rejected by Bohm and Aharonov). This hypothesis and its predicted effect on $\Delta(\Phi)$ was shown to be incorrect in subsequent literature e.g. Caradonna et. al., *Journal of Physics Comms*, 2019, Volume 3, Number 10, Page 105005]. (also see comments in previous rounds of review). The Caradonna work developed the Bohm and Aharonov theory into a matrix formalism to prove this.

The entangled nature of the photons is not something hypothesised in our paper as the referee has misinterpreted. The reviewer's misunderstanding of this system perhaps derives from the misconceptions indicated in Q3,6,9,10, and a lack of direct experience in this field. We clearly show that the established entangled theory of double Compton scattering gives agreement with the $\Delta(\Phi)$ data (fig 2). This confirms what has been observed in the previous works in more limited kinematics. The world's data cannot be explained without the entangled wavefunction. This is clear from Fig 2 where the expectation from such a hypothetical non-entangled system is also presented. Without entanglement this prediction gives the maximum $\Delta(\Phi)$ correlation achievable. If the referee was somehow correct (and the entire field has been wrong for many decades) then why don't the data (or all the previous measurements) follow this curve? The measured $\Delta(\Phi)$ correlation is clearly in agreement with the entangled theory (we note the reviewer somehow appeared unaware that the points in Fig 2 were actually data – see Q19).

6) Depending on the angle between the photons, one can guess if their state was entangled or not in origin. Then, a reasonable approach might be to assume that the photons detected in a particular angle where originated from this process. However, this reasoning has its flaws. The first one is how can you distinguish the photons that come from this interaction from the background photons, being the only difference that some were originally entangled and not the others.

As discussed above - we do not identify or distinguish. We use the predicted correlations arising from entanglement (and the differences in these correlations compared to those from backgrounds) to separate their contribution on a statistical basis. This is clearly presented in the paper. The reasoning stated by the referee about “distinguishing” photons event-by-event is indeed flawed, does not respect the probabilistic basis of quantum mechanics and we don't do it. It is not possible to separate on an event-by-event basis. However, the method we actually use is valid.

We do not get the impression the reviewer grasped what was done in the paper with much depth.

7) As stated above, the photons that arrive at the detector are not entangled, so all photons detected are indistinguishable in terms of entanglement.

This rather astounding and unjustified claim is based on flawed assumptions. Please refer to Q5. This would contradict all that is currently known about gamma interactions and the entire literature in the field.

8) Secondly, you will need also to discard (or quantify) any other possible physical process that can produce entangled photons and that can occur between the target and the detectors. Since a PET is not performed under isolated and controlled conditions, one should expect, in my opinion, other sources of entangled photons.

In the context of PET imaging, it is inconceivable that photon pairs in this energy range can be produced by any other process. We are not aware of any other process which could produce them. The referee doesn't propose any process and we have no idea what he/she conceives of. The GEANT4 simulation already simulates all known interaction processes occurring in this energy regime. The annihilation photons are at energies below the threshold to produce further e-e+ pairs so this is not a possibility. Other reactions (photoelectric effect, Rayleigh) not only give negligible contributions - they also would not produce entangled gamma photon pairs in the appropriate energy range. Gamma interactions with nuclei are negligible (and also modelled). All these processes are in the simulation and are simply not contributing at a level to affect the results. GEANT4 is the world's leading and most comprehensive simulation of particle interactions with matter in this energy range. Such "other sources" would show up in the simulation - the assertion is just speculative and in our opinion incorrect.

9) From this point, the authors misinterpreted the results of the references [1], [12,13]. First, they state "The annihilation [photon] have orthogonal linear polarisation with an entangled wave function which can be expressed as" and write eq(1) which corresponds with the single state of the two-photon linear polarizations. As I said in the previous paragraphs and as it is described in (for instance) Ref. [12] this interaction generates a pair of photons which entanglement properties depend on the relative angle between them.

In the centre-of-mass of the e+e- system the annihilation photons are emitted back-to-back. The photons are entangled and there is not a dependence of the entanglement on "angle" of the gamma pair as suggested by the referee. When the two photons are measured in a double Compton scattering process, the enhancement of the delta(phi) (between the Compton scattered photons) shows an angular dependence. This is nothing to do with the amount of entanglement changing with angle - it derives from the analysing power of the Compton scattering reaction for polarisation (which is angle dependent). The referee appears to misunderstand this as somehow reflecting differing amounts of photon entanglement with scatter angle. This is just incorrect - it is the sensitivity to polarisation in the Compton scattering that changes with angle. For the current paper we choose appropriate polar angle ranges where the sensitivity is enhanced.

10) That is, these two photons can outcome as a product state of their polarizations or as an entangled state. Therefore, it is incorrect to assume that all outgoing photons of the reaction e+e- to 2 photons are entangled because they are not. I believe that the authors may have misinterpreted this result after reading reference [1], where only the maximal entangled case is discussed.

This is incorrect. The state used in Ref [1] is the only allowed state as discussed in the response to Q5. That is why Bohm and Aharanov calculated for this state. To include others would require violations of angular momentum/parity conservation and a reinterpretation of the entire field. To somehow include these (and disregard the conservation laws) would likely lead to glaring discrepancies between the predictions of quantum mechanics and all the measured data in this field.

11) Another important point is which formula the authors are introducing to their simulations. The step from eq(1) to eq(2) is a mystery for me. It is not detailed how the authors use eq(1) to arrive at eq(2), which seems the central formula of all their analysis. The reference that they provide (Ref.[13]) is a less than a page article from 1947 with only one equation and without describing any derivation.

The authors did not “arrive” at equation 2. The use of the equation (and underlying theory) is entirely appropriate and well established in the literature. The reference [13] provides the result in the form appropriate to include the prediction in the simulation (i.e. differential cross section equation 2). In the paper we state it builds on the earlier derivations of Pryce and Ward which is also referenced (in the current paper and in Ref [13]). Snyder derives the step from equation 1 to equation 2 by two methodologies, using perturbation theory or the Klein Nishina theory. Pryce’s thesis (referenced in the papers) gives more details on the derivation.

This paper is not some new derivation of equation 2. We use the equation as it is established in the literature. We don’t think it is appropriate in an experimental paper to reproduce the theoretical derivations. We are surprised the reviewer sees this as some kind of omission despite it being referenced. The reviewers apparent lack of acceptance of the established theory in this field should not be used as part of a review of the current paper.

12) That reference states “If the two gamma-quanta emitted on the annihilation of a slowly moving positron-electron pair both undergo a Compton scattering...” and I believe that the formula provided is another form of the Klein-Nishina formula. However, this formula applies to Compton scattering and has anything to do with the possible previous entanglement between the photons involved.

Compton scattering is a polarization dependent process (photons tend to scatter in planes which are perpendicular to their polarization states). Therefore, Compton scattering yields information of the polarization correlations between photons as the Pryce-Ward cross section (equation 2) shows. The entanglement of polarisation therefore directly influences the entangled cross section for double Compton scattering - and we are astounded the referee somehow implies this is not the case, in contradiction with all work in the field.

We are of the opinion the reviewer has misunderstood the basis of the theoretical papers rather than discovering some failing in these (and many other) theorists and experimentalists work (also see response to Q13 below)

13) The authors have incorrectly assumed that somehow eq(1) (maximally entangled photons originated in one process) is included in a cross-section of a different process.

This was not “assumed” by the authors but derived by leading theoretical figures in quantum mechanics such as Bohm and Aharonov, Pryce and Ward, Snyder as well as many theoretical works up to the most recent work of Caradonna[]. Without justification the reviewer seems to question the entire theoretical basis of the field, while assuming it is some ad-hoc mixing made for the current paper. This is an incorrect and provocative comment and illustrates the reviewers lack of familiarity with the field.

14) At this point, giving the fact that the authors always talk about the “quantum-entangled photons” but not all photons from this kind of reaction are entangled, it is not clear how they can interpret the results under the initial claim “we carry out a first simulation of quantum entanglement for photons in the MeV regime”.

This speculated contribution of non-entangled photons is a misconception as discussed in our responses to questions above. We infer this arises from the reviewer's lack of familiarity with the field, which is stated in the preamble and evident in the critique. The comment seems to derive from a misunderstanding that the entanglement is angle dependent (it is not), that the wavefunction must immediately collapse (it does not) and that other non-entangled states are allowed (they are not).

The paper does present the first simulation of photon interactions in matter at the MeV scale which includes the correlations between reaction products arising from entanglement. We show this has a clear effect on the predictions and agrees with experimental data. We would be happy to change "a first simulation" to "a first simulation of the influence of quantum entanglement in photon-matter interactions in the MeV regime" to make it more specific. However we don't feel the reviewer has a good grasp of what has been achieved.

15) They are also mixing the formalism of two different processes, e^+e^- and Compton scattering.

We are astounded the referee questions this - with an assertion which appears to contradict the validity of all previous theoretical and experimental work in the field (see response to Q13. We reiterate that we did not mix anything – it was done by leading figures in the development of quantum mechanics, has been derived using a number of different methodologies and has entirely stood the test of time.

16) The next paragraphs describe the details of the PET simulation. I am not familiarized with this kind of simulations, so I cannot give an educated analysis of it. However, my concerns stand: are the authors assuming that all photons generated are entangled?

Yes – in full agreement with established theory

17) Does eq.(2) quantify properly the entanglement of these photons?

Yes – see response to Q10-15

18) The authors also mention multiple times the "entanglement effects". What are these "effects"?

We are dealing with entanglement of linear (or circular) polarisation – and the effect of entanglement on the scatter planes of double Compton scattering. The predicted effects are included in equation 2. The effect of not including entanglement is also shown by non-entangled curves in Fig 2 (from quantum mechanical theory and NOT somehow derived in this work). We should remark that such a non-entangled system from positron annihilation is forbidden due to conservation laws. The predictions from such a hypothetical system cannot describe the experimental data in Fig 2 (and all previous measurements of the enhancement in more limited kinematics).

19) Coming back to the simulation and the results shown in Fig.(2) I have a question motivated by my ignorance about this kind of simulations, and two observations. The question is: is the data real data or simulated data (black points)?

The black points are real data. This is stated in the first sentence of the caption. We do not understand how the reviewer could not understand this from the paper if read in any depth – it is a key result from the work. The text makes clear that the points are experimental data and how they were obtained. As a result the reviewer doesn't appreciate that the entangled theory

is in full agreement with the data and the non-entangled prediction (which he/she proposes as the reality) is clearly not.

20) The first observation is that the authors distinguish two models, the entangled and the standard, but as I argue above, photons coming from e+e- interaction can emerge both entangled or not, so it has no physical sense to try to analyze them separately with two different models.

As discussed above the reviewer is incorrect in the assertion the gamma can be “entangled or-not”, which we infer comes from a number of misconceptions. Basic conservation laws (e.g. momentum, parity) would need to be violated to allow non-entangled contributions speculated by the referee. We rather think it makes no “physical sense” to combine them as proposed by the referee.

21) The notion of “entanglement breaking” has no physical meaning. Entanglement is not a property that can be broken but a property that can be lost due to other interaction processes. This is maybe a terminology observation unless the authors have another meaning for that.

We agree that “breaking” is perhaps not an informative phrase (although used in previous literature). Entanglement is of course lost from the entangled-pair and dispersed to the environment. We are happy to change “breaking” to “loss”.

22) “Additionally, a first measurement of photon entanglement breaking for photon energies above the optical regime has been achieved”. This manuscript does not describe in any place an entanglement measurement protocol.

The entanglement is implicit in the modelling of the initial Compton scatter processes. The measured $\Delta(\phi)$ modulation is reproduced using the established theory for entangled photons (e.g Fig 2 and the other experimental works referenced in the paper).

The paper presents the first measurement of the $\Delta(\phi)$ correlations between 2 detected gamma (following an intermediate scattering process). This is the first measurement in the MeV regime and we feel, an important result for the field. The measured data do not agree with the hypothesis that entanglement is maintained and agree with a modelled scenario where entanglement is lost. This is the basis on which the statement is made. As mentioned in the paper improved statistical accuracy is required to achieve more detailed constraint.

In this energy regime the interaction of entangled particles with their environment is less complicated than the optical regime where noise, decoherence, short interaction path lengths and other effects mean that quantifying entanglement often necessitate strict protocols. The current work opens up programmes for future more detailed measurements. At that point more detailed protocols may be applicable – but not here.

23) Moreover, photons do not arrive entangled to the detector, so talking about entanglement detection has no physical sense here.

This is just incorrect. As discussed above the assertion is not backed up by any previous theoretical or experimental works, or even the current results (e.g. Fig 2).

24) Besides all the controversial points described above, I would like to add another one.

We agree the reviewer's comments are controversial, but disagree that there is anything controversial about the theory we adopted in the paper.

25) To my understanding, the authors aimed to prove that entangled photons play a relevant role in PET imaging and that the use of entanglement formalism may prove useful in the reconstruction of these images. However, they do not discuss nor mention the time evolution of the hypothetical photon's wavefunction. This is not a particularly problematic point if all the analysis is performed at 0 time, that is, at the generation point (formalism already developed in some references, for instance, Ref. [10]), but the goal is to use it in a real-world application, where noise and decoherence are unavoidable.

For the entangled gamma state itself (neglecting decoherence) early research on quantum mechanics and positron annihilation gamma included speculation about the transition from an entangled to a separable state. The reviewer may be unaware of works showing the constancy of the $\Delta(\Phi)$ correlation for e^-e^+ annihilation gammas with distance [e.g Wilson, J. Phys. G 2, 613 (1976)]. As shown in the literature this is not to be the case (at least up to 2 metres, well beyond the coherence length/lifetime of the positronium) [Ref]. If such effects exist (against the currently accepted interpretation of the entire field) they are not evident at distance scales appropriate for PET.

For the decoherence aspects - the reviewer is misguided in viewing decoherence as "unavoidable" in this energy range (also see Q3). If, as the reviewer speculates, somehow the photons transition to a non-entangled or unpolarised state immediately after creation (which we do not agree with) then the inability of such a state to explain the new data in Fig 2 (and all previous works) would indicate a fundamental failing of quantum mechanics. This is clearly not the case.

26) For all the reasons stated above, I conclude that this work is ill-defined (uses well-known formalism incorrectly, for instance, by mixing Compton scattering cross-sections with entangled wave functions from a different process without any intermediate justification),

This "mixing" was done by leading theorists in the development of quantum mechanics. The reviewer assumes it was done by the authors which is incorrect. It is fully available in the literature. The works are justified and tested over decades.

We do not think the reviewer has any valid basis to criticise the wealth of previous theoretical work – it is clear he/she does not have a background in entanglement at these energy scales.

27) the hypothesis is physically incorrect (photons that arrive at the detector are not entangled, so an entanglement analysis there has no physical sense)

As discussed above this is a clear misunderstanding of gamma interactions in matter by the reviewer, probably guided by experience in the optical regime. This comment is entirely inconsistent with the body of theory in the field and all published experimental results.

28) and, therefore, the conclusions cannot be interpreted under the hypothesis assumptions. I will not discard that a proper analysis of entanglement in processes like e^+e^- and its correct insertion in the PET simulations, might be scientifically relevant, but I am afraid that this work does not present the necessary formalism.

We reiterate that the formalism we use is the established quantum mechanics and that the speculative formalisms proposed by the reviewer (e.g. mixed non-entangled and entangled contributions) are forbidden by conservation laws and would have no prospect of describing the experimental data (in the current paper or previous works)

Reviewers' Comments:

Reviewer #4:

Remarks to the Author:

Dear editor and authors,

The new manuscript version, together with the detailed response of the authors, has addressed my comments and concerns about the previous version of this work. The explanations and references added have clarified some obscure points, in my opinion, and some misconceptions that I had when I read the previous version.

The current version fulfills the requirements for publication.

Reviewer #5:

Remarks to the Author:

I have read the manuscript and the most recent round of review including the authors' response and have come to the following conclusion:

1) First of all, here is my summary of what the manuscript claims:

- It discusses whether photon entanglement can be beneficial for PET imaging
- It experimentally studies the correlation in the polarization of the photon pair emitted in positron annihilation events. For the case without any scatterer in the photon path, it claims the presence of entanglement. However, this entanglement has been theoretically predicted and experimentally demonstrated previously in the literature.
- It performs an analogous measurement with a scatterer in the path of one photon, to simulate a patient
- It reports an extension of the widely used Geant4 package to include the effects of entanglement expected to be relevant to the present experimental setup, and claims that the modelling achieved by this is consistent with the experimental data with and without scatterer in the photon pathway
- It then uses this new software to model a relevant setup involving a phantom to estimate the effect of entanglement in a realistic setting. As a straightforward example, it is claimed that the correlation between the photons can be used to suppress unwanted background, thereby improving the imaging.

In my view, delivering on these claims certainly warrants a publication in Nature Communications.

2) However, the referees question several of these claims. In the following, I would like to comment on some of these criticisms raised by the reviewers

- Does the manuscript demonstrate entanglement or not?

In my view, yes. It does so by showing that in the test experiment without scatterer (Fig 2) the amplitude of the modulations exceeds an "R" value beyond that allowed from "classical" theories (see around line 80). Note that previous works also demonstrated this violation of "classical bounds". This is the entanglement detection protocol asked for by reviewer 4. (The manuscript does not demonstrate entanglement in the experiment with the scatterer, nor in the simulations with the phantom, but it also does not claim it).

However, in some parts the manuscript gave me the impression that the authors imply that the agreement of their simulation (including the entanglement effects) with the data also acts as a verification of the presence of entanglement. One example is the sentence "The results illustrate the need for an account of quantum entanglement in PET simulation to correctly describe the

observed distributions."

starting in line 162. From a formal point of view, this is definitively not the case, because it cannot be ruled out that some other mechanisms unaccounted for in the modelling lead to the experimental observations. The results only show that the restricted code (red/green line) cannot describe the measurement. I understand that it is tempting to infer entanglement from the agreement with the modeling, but it is simply not justified. The manuscript should be carefully checked for such potentially misleading formulations.

- Does entanglement play a role / what is the effect of decoherence?

Referee 4 states "How can two photons (entangled or not in origin) hold the entanglement properties through a human body and be detected cm away? This is physically impossible, and the photons that arrive at the detector are surely not entangled. Their wave function collapses almost instantaneously after the pair is generated due to the noise and decoherence."

Referee 2 also has questions related to the effect and the modelling of the entanglement.

Lets consider two entangled photons emitted in an annihilation process in the center of a human body. If they leave the body and arrive at the detector without any interaction with the environment, then certainly they remain entangled. The referee's statement therefore can be made more precise by asking: What is the probability for such events without any interactions of the entangled photon pair with the environment? The referee's statement implies that this probability is negligibly small. I would expect the same for visible photons, but the authors consider high-energy photons. They measure this probability (at least indirectly in the test experiment with the scatterer) and they also calculate it. This in essence is what Geant4 does: It contains probabilities for all relevant processes (i.e., interactions) and applies them using a Monte Carlo approach.

It is not easy to understand from the presentation how the decoherence is included in the simulation, even though the authors comment on it in various places. I understand it as follows. Initially, there is the polarization-entangled photon pair. If this pair undergoes a double-compton scattering, then the outcome (angular distribution, cross section etc) will be described by Eq. (2). It is further assumed that this initial interaction "decoheres" the wave function. Consequently, the subsequent dynamics of these photons is calculated using the standard Geant4 routines not assuming entanglement. In particular, further scattering events of the two photons are no longer correlated as given by Eq. (2). In the ideal case, the single scattering event occurs in the CZT crystals, which is required to measure the angular distribution and thereby study the entanglement properties. If the initial scattering occurs in the scatterer/phantom/patient, then the photons arriving at the CZT detectors are assumed separable.

To resolve this issue, I would recommend the following: The authors should be able to estimate the fraction of entangled photons which leave the phantom without any further interaction until they reach the detector crystals from the simulations, where they undergo the DCS. I would encourage them to discuss this fraction in the paper. This number is of primary interest already independent of how the actual entanglement dynamics is implemented in the code. One could even think about defining a "scatter-free pathlength" to be able to estimate the influence of depth in the patient and the type of material in the photon pathway.

I would also encourage the authors to better describe the way the entangled photon pair is included in the code. I could imagine that a simple flow diagram would be very helpful. The top box could be the initially entangled photon pair. It has different options in the MC simulation: No scattering, one DCS scattering, one DCS and further subsequent scatterings, and maybe also other non-DCS processes. These options could be represented by branches originating from the top box in the diagram. In each branch, it could be indicated which interaction is modelled using which theory (i.e., Eq. (2) or the standard Geant routines). One could even make this diagram more useful by separating the different branches further into scattering in the phantom/patient and/or scattering in the CZT detector. This would lead to more branches, but would allow the authors to indicate which branches are favorable for the PET imaging or not.

- Code availability

Details on the implementation of the code are not included in the manuscript, but the code has been made available to the reviewers and a publication within the Geant4 package is anticipated. I am satisfied with this.

- Inconsistencies between experiment and simulation.

In my view, the agreement between experiment and simulation is good. One should not forget that even though the concept of the experiment is simple, the actual implementation is not, due to a number of "real life" limitations. Geant4 is an established tool to take these into account. These "real life" effects are expected to lead to corrections to the naive expectations for the idealized setup, as discussed in the authors' response.

- Claims of novelty in the manuscript.

I cannot understand the reviewers' remarks in this direction. In my view, the present form of the manuscript clearly states what is new and what is not. This in particular includes the discussion of entanglement effects in the annihilation photons which has a history of more than 70 year. I would further remark that the abstract and title do not claim the demonstration of entanglement itself as a result of the paper, consistent with the observation that this result is not new. For the purpose of the present paper, it is the reduction of the entanglement with scattering events in the patient/phantom/... which is of relevance, and therefore mentioned in the abstract.

- "overselling" by the notion "quantum entanglement"

The topic of the manuscript is the study of whether quantum entanglement is eneficial for PET or not. I do not see any way of studying this question without mentioning it. The authors experimentally demonstrate that entanglement exists without patient/phantom/scatterer, consistent with previous analogous demonstrations. The usefulness of (initially) entangled photons is then convincingly demonstrated using simulations. I therefore do not see any overselling

- spatial/polarization entanglement, assumptions leading to (1):

The manuscript is fully clear about what types of correlations / entanglement is discussed. I would like to remark that the original source of entanglement captured in Eq. (1) is two-photon annihilation of a single particle with known quantum numbers. This process and the properties of the outgoing photons are well-studied in the literature. The properties in essence follow from conservation laws (most importantly, momentum and (total) angular momentum), on the level of introductory physics courses. The step from (1) to (2) is less obvious, but similar results have been obtained or used in a number of references, and not only "very old ones" (I also do not see why older references should be less good than younger ones)

3) Summary

In summary, I believe that the claims mentioned above are satisfied in the manuscript. However, there are some shortcomings in the presentation mentioned above. In my view, they are (at least part of) the reasons for some of the referees' comments. I therefore recommend publication after corresponding manuscript revisions have been applied.

Response to reviewer's comments

The reviewer comments are reproduced in blue text and our responses are shown in green text. Changes to the manuscript arising from the comments (where applicable) are quoted in Brown text.

Reviewer #4 (Remarks to the Author):

Dear editor and authors, The new manuscript version, together with the detailed response of the authors, has addressed my comments and concerns about the previous version of this work. The explanations and references added have clarified some obscure points, in my opinion, and some misconceptions that I had when I read the previous version.

The current version fulfills the requirements for publication.

We are happy the amendments to the manuscript made the case more clearly. We thank the reviewer for the comments and feel the readability of the manuscript for a general audience has been improved.

Reviewer #5 (Remarks to the Author):

I have read the manuscript and the most recent round of review including the authors' response and have come to the following conclusion:

1) First of all, here is my summary of what the manuscript claims:

- It discusses whether photon entanglement can be beneficial for PET imaging
- It experimentally studies the correlation in the polarization of the photon pair emitted in positron annihilation events. For the case without any scatterer in the photon path, it claims the presence of entanglement. However, this entanglement has been theoretically predicted and experimentally demonstrated previously in the literature.
- It performs an analogous measurement with a scatterer in the path of one photon, to simulate a patient
- It reports an extension of the widely used Geant4 package to include the effects of entanglement expected to be relevant to the present experimental setup, and claims that the modelling achieved by this is consistent with the experimental data with and without scatterer in the photon pathway
- It then uses this new software to model a relevant setup involving a phantom to estimate the effect of entanglement in a realistic setting. As a straightforward example, it is claimed that the correlation between the photons can be used to suppress unwanted background, thereby improving the imaging.

In my view, delivering on these claims certainly warrants a publication in Nature Communications.

We are happy the reviewer supports the publication of our work in Nature Comms.

2) However, the referees question several of these claims. In the following, I would like to comment on some of these criticisms raised by the reviewers

- Does the manuscript demonstrate entanglement or not?

In my view, yes. It does so by showing that in the test experiment without scatterer (Fig 2) the amplitude of the modulations procudes an "R" value beyond that allowed from "classical" theories (see around line 80). Note that previous works also demonstrated this violation of "classical bounds". This is the entanglement detection protocol asked for by reviewer 4. (The manuscript does not demonstrate entanglement in the experiment with the scatterer, nor in the simulations with the phantom, but it also does not claim it).

However, in some parts the manuscript gave me the impression that the authors imply that the agreement of their simulation (including the entanglement effects) with the data also acts as a verification of the presence of entanglement. One example is the sentence "The results illustrate the need for an account of quantum entanglement in PET simulation to correctly describe the observed distributions." starting in line 162. From a formal point of view, this is definitively not the case, because it cannot be ruled out that some other mechanisms unaccounted for in the modelling lead to the experimental observations. The results only show that the restricted code (red/green line) cannot describe the measurement. I understand that it is tempting to infer entanglement from the agreement with the modeling, but it is simply not justified. The manuscript should be carefully checked for such potentially misleading formulations.

We agree that the detailed agreement of the experimental data with the entangled theory does not absolutely preclude another (currently unknown) mechanism providing the measured enhancements beyond the limit for a non-entangled state. We have checked the formulation of the text is compatible with the reviewer's comments. The modifications are listed below:

Line 134 (replacing the phrasing highlighted by the reviewer above):

The results in Fig 2, and previous measurements in more limited kinematics from a range of different Pa sources [2, 10, 12, 17–24], show that the azimuthal correlation of the Compton scatter planes in DCSc of Pa photons is in agreement with the entangled theory (equation 2) and has a correlation (R) beyond the upper limit of a separable non-entangled state

Line 3: quantum entanglement -> predicted quantum entanglement

Line 23: described by-> predicted to be in a

Line 84 validated->tested

Line 96 validated->tested

Line 114: show a good agreement with the measured $\Delta\varphi$ distribution ($\chi^2/\nu = 1.87$),

providing validation of the QE-Geant4 simulation and confirming the entangled theory is consistent with experimental data (the agreement on a bin-by-bin basis is presented in Supplementary Note 4)

Line 163: The ability of the new QE-Geant4 to accurately describe the observed correlations in DCSc offers new possibilities to separate the true (assumed entangled) PET events from backgrounds of scatter and random events.

Line 236: gave a good description of the measured correlation between the Compton scatter planes, while predictions based on a (hypothetical) non-entangled state could not describe this correlation

Line 467: the quantum entanglement -> the predicted quantum entanglement

Line 483: the effects of quantum entanglement -> the predicted effects of quantum entanglement

Line 486: the entanglement -> the predicted entanglement

Supp Line 6: In this note we show analysis which confirms that the simulation outputs in the new QE-Geant4 simulation, are in agreement with the analytical entangled theory forming the basis of the simulation (i.e. equation 2 in the main paper). We also show our analysis which confirms that the standard Geant4 is in agreement with the analytical theory predictions [1] for a (hypothetical) non-entangled state.

We also added clearer links to the Supplementary notes at appropriate points in the text (lines 80, 122, 274).

- Does entanglement play a role / what is the effect of decoherence?

Referee 4 states "How can two photons (entangled or not in origin) hold the entanglement properties through a human body and be detected cm away? This is physically impossible, and the photons that arrive at the detector are surely not entangled. Their wave function collapses almost instantaneously after the pair is generated due to the noise and decoherence." Referee 2 also has questions related to the effect and the modelling of the entanglement. Lets consider two entangled photons emitted in an annihilation process in the center of a human body. If they leave the body and arrive at the detector

without any interaction with the environment, then certainly they remain entangled. The referee's statement therefore can be made more precise by asking: What is the probability for such events without any interactions of the entangled photon pair with the environment? The referee's statement implies that this probability is negligibly small. I would expect the same for visible photons, but the authors consider high-energy photons. They measure this probability (at least indirectly in the test experiment with the scatterer) and they also calculate it. This in essence is what Geant4 does: It contains probabilities for all relevant processes (i.e., interactions) and applies them using a Monte Carlo approach.

It is not easy to understand from the presentation how the decoherence is included in the simulation, even though the authors comment on it in various places. I understand it as follows. Initially, there is the polarization-entangled photon pair. If this pair undergoes a double-compton scattering, then the outcome (angular distribution, cross section etc) will be described by Eq. (2). It is further assumed that this initial interaction "decoheres" the wave function. Consequently, the subsequent dynamics of these photons is calculated using the standard Geant4 routines not assuming entanglement. In particular, further scattering events of the two photons are no longer correlated as given by Eq. (2). In the ideal case, the single scattering event occurs in the CZT crystals, which is required to measure the angular distribution and thereby study the entanglement properties. If the initial scattering occurs in the scatterer/phantom/patient, then the photons arriving at the CZT detectors are assumed separable.

This is indeed the procedure we followed.

To resolve this issue, I would recommend the following: The authors should be able to estimate the fraction of entangled photons which leave the phantom without any further interaction until they reach the detector crystals from the simulations, where they undergo the DCS. I would encourage them to discuss this fraction in the paper. This number is of primary interest already independent of how the actual entanglement dynamics is implemented in the code. One could even think about defining a "scatter-free pathlength" to be able to estimate the influence of depth in the patient and the type of material in the photon pathway.

We have now included quantification of the scatter fractions for the simulated preclinical scanner with NEMA phantom (lines 227-231; text reproduced below).

The PET study presented here corresponds to a preclinical PET scanner with a small-animal equivalent phantom. The scatter probability from the small-animal phantom, obtained from the QE-Geant4 simulation, is 15% providing a test of the QE-PET technique for the challenging scenario of small scatter backgrounds. Human PET, as referenced in the introduction, provides larger scatter contributions of up to 67% and should be more amenable to the method proposed.

We now quote the typical scatter probabilities in pre-clinical (NEMA_NU4 phantom used in the current study - extracted from the simulation) and human PET (derived from the scatter to true ratios quoted on lines 37-38).

The suggestion of the reviewer to extract a scatter-free pathlength is nice, but as in PET there are so many permutations of medium traversed, path length for different scan types, its description and presentation may become rather involved. We feel the amendments above give a sense of the typical scales of scatter probability for the reader. We hope this satisfies the reviewer's comments.

I would also encourage the authors to better describe the way the entangled photon pair is included in the code. I could imagine that a simple flow diagram would be very helpful. The top box could be the initially entangled photon pair. It has different options in the MC simulation: No scattering, one DCS scattering, one DCS and further subsequent scatterings, and maybe also other non-DCS processes.

These options could be represented by branches originating from the top box in the diagram. In each branch, it could be indicated which interaction is modelled using which theory (i.e., Eq. (2) or the standard Geant4 routines). One could even make this diagram more useful by separating the different branches further into scattering in the phantom/patient and/or scattering in the CZT detector. This would lead to more branches, but would allow the authors to indicate which branches are favorable for the PET imaging or not.

We have now structured the paper so that the implementation of entanglement in the simulation is discussed more fully in one complete section in the methods (lines 253-276). We agree a flowchart is informative and we have now included this (Supplementary figure 6)

Code availability. Details on the implementation of the code are not included in the manuscript, but the code has been made available to the reviewers and a publication within the Geant4 package is anticipated. I am satisfied with this.

We are happy the reviewer sees our code availability plan as appropriate.

We have added additional text in the methods stating which version of Geant4 was used for the studies in the paper (line 257)

Inconsistencies between experiment and simulation. In my view, the agreement between experiment and simulation is good. One should not forget that even though the concept of the experiment is simple, the actual implementation is not, due to a number of "real life" limitations. Geant4 is an established tool to take these into account. These "real life" effects are expected to lead to corrections to the naive expectations for the idealized setup, as discussed in the authors' response.

We agree - and view this first combination of the predictions of entanglement with detailed simulation as one of the important new results from the paper. We are happy the reviewer recognises this. We have added an additional line to highlight how the simulation development offers further opportunities.

Line 241: Additionally, the framework offers possibilities for further fundamental tests of entanglement at the MeV scale

-Claims of novelty in the manuscript: I cannot understand the reviewers' remarks in this direction. In my view, the present form of the manuscript clearly states what is new and what is not. This in particular includes the discussion of entanglement effects in the annihilation photons which has a history of more than 70 year. I would further remark that the abstract and title do not claim the demonstration of entanglement itself as a result of the paper, consistent with the observation that this result is not new. For the purpose of the present paper, it is the reduction of the entanglement with scattering events in the patient/phantom/... which is of relevance, and therefore mentioned in the abstract.

- "overselling" by the notion "quantum entanglement"

The topic of the manuscript is the study of whether quantum entanglement is eneficial for PET or not. I do not see any way of studying this question without mentioning it. The authors experimentally demonstrate that entanglement exists without patient/phantom/scatterer, consistent with previous analogous demonstrations. The usefulness of (initially) entangled photons is then convincingly demonstrated using simulations. I therefore do not see any overselling

- spatial/polarization entanglement, assumptions leading to (1):

The manuscript is fully clear about what types of correlations / entanglement is discussed. I would like to remark that the original source of entanglement captured in Eq. (1) is two-photon annihilation of a single particle with known quantum numbers. This process and the properties of the outgoing photons are well-studied in the literature. The properties in essence follow from conservation laws (most importantly, momentum and (total) angular momentum), on the level of introductory physics courses. The step from (1) to (2) is less obvious, but similar results have been obtained or used in a number of

references, and not only "very old ones" (I also do not see why older references should be less good than younger ones)

3) Summary

In summary, I believe that the claims mentioned above are satisfied in the manuscript. However, there are some shortcomings in the presentation mentioned above. In my view, they are (at least part of) the reasons for some of the referees' comments. I therefore recommend publication after corresponding manuscript revisions have been applied.

We hope the changes outlined above have addressed all the reviewer's comments on presentation and we thank the reviewer for the useful comments which have improved the manuscript.

We comment that in the revised version the format and structure of the paper has been changed to better match the Nature Comms guidelines - as recommended by the editor. A summary of these changes is listed in Appendix 1 below.

Yours sincerely,
Daniel Watts (for the authors)

Appendix 1: Changes to manuscript structure in response to Editors comments:

To comply with our article templates, the text must be split into:

- Introduction (ideally 1000 words or less), which must include the background and rationale for the work. The final paragraph should be a brief summary of the major results and conclusions. The results of the current study should only be discussed in this final paragraph. We reduced the word count to 1095 in the revised manuscript. We hope this is within the acceptable length. We have moved some of the discussion about detailed aspects of the previous Positron annihilation measurements (e.g. Bell's tests) and theory into Supplementary Note 3, referenced from the main text (line 79). However we keep the key points in the main text.
- Results, which must be split into subheaded sections, ensuring that the subheadings are no longer than 60 characters including spaces. Done
- Discussion, without subheadings. Done
- Methods, which must be split into subheaded sections, ensuring that the subheadings are no longer than 60 characters including spaces. There is no word limit for this section. Done

To achieve the above we have reordered some of the content so the paper flows as introduction, results then methods - as recommended in Nature Comms guidelines and adopted for most previous articles. This required some minor grammatical changes and additional referencing in the text, which we hope are self evident in the revised manuscript.

The whole article (excluding methods and abstract) should ideally not be longer than 5000 words. This has been done.

Figure Guidelines:

1. All figures should have a title briefly describing the whole figure. Figure titles should ideally be no longer than about one line, with minimal symbols and no punctuation. It would be ideal if you could limit the use of acronyms in figure titles. Any acronym used should be defined in the caption. This has now been done

2. Panels should be individually labelled and referred to in the caption. Do not refer to panels via their position, as this may change in the production of the final pdf. We didn't see any instances of this in the manuscript. The multipanel figures are labelled as a,b,c etc.

3. We have a 350 words limit on figure captions, therefore feel free to expand them to suitably and comprehensively describe what the readers are looking at. Ideally, figures should be as self-consistent as possible, without the need to refer to the text to grasp their meaning. The captions have been expanded to give more comprehensive discussion of the plots.

4. Please supply figures so that every element of each figure is editable (i.e. we can highlight and edit the text, and move individual parts of the figures around). When making these changes please ensure resolution stays high at 300dpi. This has now been done

5. The meaning of all error bars (sd? Sem?) and how they were calculated should be described within the captions of all figures in which they occur. This has now been done.

6. Please ensure that all plotted variables are accompanied by units of measures, unless dimensionless. Please use (a.u.) if arbitrary. We did this (with the exception of the ratio figure - Supplementary Fig 1) where the plotted data on the y-axis is dimensionless.

Supplementary Information Guidelines:

Supplementary Information should be provided as a separate pdf file. The first page should be a cover page containing only the title in the form "Supplementary Information - Title of the Manuscript" and the authors in the form "Smith et al.", with no affiliations.

This has now been done

The text in the Supplementary Information should be organised using the following subheaded sections:

- Supplementary Notes, headed "Supplementary Note 1 - *title*", "Supplementary Note 2 - *title*", etc.
- Supplementary Discussion
- Supplementary Methods
- Supplementary References, which should be self-contained. That is, references mentioned in both the main text and the Supplementary Information should be part of both reference lists so that the Supplementary Information does not refer to the reference list in the main paper and vice versa. References cited in the Supplementary Information should be numbered sequentially from 1. They should be formatted in the same style as the main paper. This has all now been done

We also removed all footnotes (as per the guidelines). The previous footnote text is now included in the text of the main paper or in the supplementary materials.

We also included an additional picture of the experimental setup in the methods (Fig 8) to add more context for the reader and hope this is acceptable.

Reviewers' Comments:

Reviewer #5:

Remarks to the Author:

The reply and revised manuscript resolve all issues raised in my initial report, also for the readers. The improved discussion and the added flow diagram are very helpful in my view. I therefore recommend publication in Nature Communications.

Reviewer#5 (remarks to the author)

The reply and revised manuscript resolve all issues raised in my initial report, also for the readers. The improved discussion and the added flow diagram are very helpful in my view. I therefore recommend publication in Nature Communications.

We are glad the modifications are acceptable and thank the reviewer for the very useful comments to improve the clarity of the paper